

# Surveillance Camera-Based Deep Learning Framework for High-Resolution Ground Hydrometeor Phase Observation

Xing WANG[1,2,3], Kun ZHAO[2,3], Hao HUANG[2,3], Ang ZHOU[2,3], Haiqin CHEN[2,3]

[1]School of Computer Engineering, Nanjing Institute of Technology, Nanjing, 211167, China

[2]School of Atmospheric Sciences, Nanjing University, Jiangsu 210023, China

[3]Key Laboratory of Radar Meteorology and State Key Laboratory of Severe Weather, China Meteorology Administration, Beijing 100044, China

*Correspondence to*: Kun ZHAO (zhaokun@nju.edu.cn)

**Abstract.**

Urban surveillance cameras offer a valuable resource for high spatiotemporal resolution observations of ground hydrometeor phase (GHP), with significant implications for sectors such as transportation, agriculture, and meteorology. However, distinguishing between common GHPs—rain, snow, and graupel—present considerable challenges due to their visual similarities in surveillance videos. This study addresses these challenges by analyzing both daytime and nighttime videos, leveraging meteorological, optical, and imaging principles to identify distinguishing features for each GHP. Considering

both computational accuracy and efficiency, a new deep learning framework is proposed. It leverages transfer learning with a pre-trained MobileNet V2 for spatial feature extraction and incorporates a Gated Recurrent Unit network to model temporal dependencies between video frames. Using the newly developed 94-hour Hydrometeor Phase Surveillance Video (HSV) dataset, the proposed model is trained and evaluated alongside 24 comparative algorithms. Results show that our proposed method achieves an accuracy of 0.9677 on the HSV dataset, outperforming all other relevant algorithms. Furthermore, in

real-world experiments, the proposed model achieves an accuracy of 0.9301, as validated against manually corrected Two-Dimensional Video Disdrometer measurements. It remains robust against variations in camera parameters, maintaining consistent performance in both daytime and nighttime conditions, and demonstrates wind resistance with satisfactory results when wind speeds are below 5 m/s. These findings highlight the model's suitability for large-scale, practical deployment in urban environments. Overall, this study demonstrates the feasibility of using low-cost surveillance cameras to build an

efficient GHP monitoring network, potentially enhancing urban precipitation observation capabilities in a cost-effective manner.

## 1 Introduction

Hydrometeors refer to any atmospheric particle consisting of liquid or solid water, which are integral to precipitation processes and play a crucial role in the water cycle and cloud microphysics (Pruppacher, Klett and Wang 1998). The





identification of ground hydrometeor phase (GHP) contributes to the improvement of quantitative precipitation estimation algorithms and promotes the understanding of precipitation microphysical processes, thus providing scientific support for the improvement of microphysics scheme of numerical weather prediction. Common examples of ground hydrometeors include rain, snow, graupel. They account for more than 90% of the GHP and influence urban transportation, communication, electricity, and other industries (Casellas et al. 2021b, Zhou et al. 2020). Especially in winter, a weather process may contain

multiple hydrometeor phases and often co-exist or convert to each other. Given the same amount of precipitation, the impacts of different hydrometeor phases may vary considerably (Leroux, Vionnet and Thériault 2023). For example, if 5 mm of precipitation falls in 24 hours, it is only light rain for the rainfall phase but heavy snow for the solid snowfall, severely influencing social production and life. In winter, when snow, rain, and graupel co-exist or interchange frequently, it is tough for forecasters to know the actual weather conditions, which seriously affects the quality of forecasts (Haberlie, Ashley and

Pingel 2015). Therefore, the accurate discrimination of the GHP, especially for the rain, snow, and graupel, has significant scientific and practical value.

Nowadays, many countries and regions no longer observe precipitation information manually (e.g., in January 2014, China cancelled ground-based manual observation). Ground-based Disdrometers (i.e., OTT Parsivel, Two-Dimensional Video Disdrometer), airborne optical and electromagnetic wave detection devices, and dual-polarization radars have become the

primary tools (Jennings et al. 2023). However, (1) with the rapid development of urbanization, ground-based disdrometers in urban areas face outstanding problems such as high construction costs, difficulty in management and maintenance, and low deployment density, resulting in limited spatial representativeness of the GHP observations (Arienzo, Collins and Jennings 2021). (2) The data collected by airborne equipment is mainly used for validating and analysing scientific experiments, which is challenging to apply on a large scale and in real-time observation tasks (Schirmacher et al. 2024). (3) Dual

polarization radar can alternatively or simultaneously transmit and receive polarized waves in both horizontal and vertical directions to obtain the echo information in different directions of the target scatterer and thus identify the hydrometeor phase in the cloud (Casellas et al. 2021a). However, precipitation particles undergo a complex physical evolution from high altitude to the ground, especially in urban areas, where the temperature may have significant spatial differences, leading to large differences in the hydrometeor phase between regions (Speirs, Gabella and Berne 2017). In summary, existing

techniques have not effectively addressed GHP's high temporal and spatial resolution discrimination.

The development of the new observation method has received much attention. Some researchers have adopted the idea of "citizen science" by encouraging residents to report precipitation they see to provide the actual value of the GHP (Crimmins and Posthumus 2022). Extensive investigations have demonstrated the effectiveness of the citizen science-based approach (Arienzo et al. 2021, Jennings et al. 2023), which provides important insights for our study. According to a survey by

Comparitech (https://www.comparitech.com/), there are approximately 770 million surveillance cameras worldwide. Surveillance video allows 24/7 observation of the precipitation process and provides clues for GHP discrimination (Wang et al. 2023a). If every surveillance camera is regarded as an observation site, such a vast number of surveillance cameras provide a high spatial resolution observation. At the same time, the surveillance video is transmitted through fiber optic, 4G,



and 5G communication networks, enabling transmission back 15-25 surveillance images per second, which offer a high
temporal resolution sensing of GHP. Moreover, the GHP observation mission can be deployed on existing surveillance
resources, showcasing the advantage of low operation and maintenance costs. Compared to the other citizen science-based
approach, the GHP observation network composed of surveillance cameras offered a more objective record of the
precipitation process, which has the potential advantages of low cost, all-day, and high spatiotemporal resolution.

However, extensive analysis and comparison experiments have revealed that rain, snow, and graupel show greater similarity
in surveillance videos (e.g., graupel and rain, as well as graupel and snow, are more similar in daytime surveillance videos
under different precipitation intensities, whereas at nighttime, the distinguishing image features of the three are much closer,
making the distinction much more difficult (For more details, see Section 3.1). This study focuses on the discrimination of
three GHP, i.e., rain, snow, and graupel particles via surveillance video, and develops a deep learning-based GHP
discrimination method. Considering that surveillance cameras capture visible and near-infrared video during daytime and
nighttime during daytime and nighttime, respectively, this study first analyzes the video imaging model of three different
particles and compares their differences in surveillance video features. Taking the above findings as a priori knowledge, a
deep learning-based GHP classification model is proposed. An efficient convolutional neural network (CNN) called
MobileNet V2 is used to extract spatial features from surveillance images based on transfer learning. These features are
stacked together and fed to a gated recurrent unit (GRU) network, which enables modeling the long-term dynamics of the
hydrometeor phase in a video sequence. Then, a hydrometeor surveillance video (HSV) dataset is constructed for the deep
learning model training and testing. Finally, the effectiveness of the proposed method is evaluated on both the HSV dataset
and the real-world experiments. To the authors' knowledge, this is the first study on graupel observation from surveillance
video data. The research findings can provide technical and data support for understanding the microphysical process of
precipitation, improving the microphysical calculation model of precipitation and improving the accuracy of satellite/radar
retrievals.

The rest of this paper is organized as follows: Following this introduction, we present the related works in Section 2; and
explain the details of the proposed deep-learning model in Section 3; and finally, we discuss the experimental results in
Section 4 and conclude in Section 5.

## 2 Literature Review

Visual perception is an effective way to distinguish GHP. Visual sensors, such as surveillance cameras, cell phones, digital
cameras, and vehicle cameras, are considered potential weather phenomena observers in existing studies. Considering the
research theme, visual is primarily defined as optical images obtained from the ground. This does not include data obtained
from LiDAR, Radar, or similar technologies. The authors divided the existing visual-based GHP identification work into
three categories: traffic surveillance cameras, in-vehicle cameras, and generalized visual data.


It should be noted that "weather" is a more generic and broad expression that includes rain, snow, haze, fog, sunny, etc. Meanwhile, "hydrometeor" mainly refers to rain, snow, graupel, or hail and is primarily attractive to professional meteorological researchers. Existing studies prefer "weather" rather than "hydrometeor."

## 2.1. Traffic surveillance camera

During rainy and snowy weather, roads suffer from snow, ice, and ponding, affecting transportation efficiency. Timely
weather information reports are significant for traffic warning, diversion, and management. However, with the limited number of weather stations and delays in radar/ satellite-based weather information release, some researchers exploited weather recognition from outdoor road surveillance cameras (Lu et al. 2014, Li, Kong and Xia 2014). As shown in Fig.1, these studies include two categories:

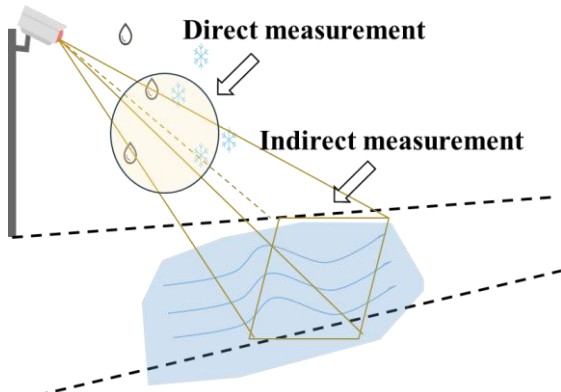

**Figure 1: Weather classification using traffic surveillance cameras.**

(1) Indirect measurement mainly refers to detecting snow, pounding, and road surface wetness from video to deduce the weather. Therefore, these methods mainly focus on the information on the road surface. For example, Shibata et al. (2014) used the texture features of the surveillance images to quantify the pattern and texture of the road surface and detect road surface conditions (wet and snow) by surveillance cameras from day to night (or low-light scenarios). Morris and Yang
(2021) constructed a road extract method by Mask R-CNN and then built a gradient-boosting ensemble classifier to predict pavement wetness. Ramanna et al. (2021) used deep CNNs to labeled the road surveillance images into five conditions and constructed a dataset for deep learning models training. Extensive experiments have shown that the EfficientNet-B4 network-based system achieved optimal performance. Landry and Akhloufi (2022) utilized the SVM and CNN to extract snow areas in the image. They built a model using surveillance cameras to estimate the percentage of the snow-covered road
surfaces. To reduce the difficulty of model training and improve accuracy, Khan, Ahmed and technology (2022) introduced a transfer learning method to apply several pre-trained CNN models for weather and road condition classification tasks. Lü et al. (2023) pre-processed the surveillance images through the road segmentation network to obtain the binary images to obtain the road image features. Subsequently, a convolutional neural network, composed of overall network branches and road network branches, was established and used to extract the overall image area features and focus on extracting the road





weather features, respectively. Askbom (2023) first used CNN-based deep learning to determine the weather condition (mainly focusing on snow), then constructed a road condition classification deep learning network with the premise of road area has been extracted by the U-Net model. Additionally, in an innovative and impressive work, Carrillo and Crowley (2020) integrated roadside surveillance images and weather data from weather stations to improve the performance of road surface condition evaluation. The fusion of surveillance cameras and other observations provides novel insights for road

weather identification.

However, there is still a period between the occurrence of rain or snow and the appearance of ponding or snow on the road surface. Thus, these Indirect methods have the disadvantage of needing to be quicker. Moreover, the above methods will not work for those surveillance cameras with no road surface or other specific region as a reference in the observed area.

(2) Direct measurement refers to identifying the hydrometeor phase by the captured information of falling raindrops,

snowflakes, or haze. For example, Zhao et al. (2011) classified weather conditions into steady, dynamic, and nonstationary and employed four direction templates to analyze the max directional length of motion blur caused by rain streaks or snowflakes. In this way, rain and snow can be distinguished from traffic cameras. After analysing the image features of different weather, Li et al. (2014) adopted the decision tree to model the image features captured during different weather conditions and built an SVM classifier to predict the weather. Afterward, Lee (2017) proposed a more straightforward

method, which used the histogram features of road images as metrics to discriminate fog and snow in road surveillance cameras. A serious CNN-based weather classification effort has been implemented with the development of a deep learning algorithm. Xia et al. (2020) took the residual network ResNet50 as the basis and proposed a simplified model for weather feature extraction and recognition on traffic roads. Sun et al. (2020) built a deeply supervised CNN to identify road weather conditions through the road surveillance system. Dahmane et al. (2021) constructed a deep CNN to differentiate between

five weather conditions from traffic surveillance cameras, such as no precipitation, foggy, and rainy. Some advanced deep learning networks or methods like Attention and transformer were naturally introduced, such as Dahmane et al. (2018), who used CNN to identify rain, fog, and snow weather from road cameras and applied on large-scale from day to night through the learning transfer method. Wang et al. (2023b) built a Multi-Stream Attention-aware Convolutional Neural network to identify sand and dust storm from city surveillance cameras. Chen et al. (2023) built a deep learning model that employs

multiple convolutional layers to extract features from weather images and a Transformer encoder to calculate the probability of each weather condition based on these extracted features.

Compared to indirect measurements, direct measurements do not require road surface conditions as a reference and thus have a broader range of applications. That is, direct measure methods can also be deployed in non-traffic surveillance cameras, which are also adopted in this study.

**2.2. In-vehicle cameras**

Some researchers concentrated on recognizing weather conditions from images captured by in-vehicle cameras. For example, Kurihata et al. (2005) used image features from PCA to detect raindrops on a windshield and to judge rainy weather. Roser



and Moosmann (2008) presented an approach that employed SVM to distinguish between multiple weather situations based on the classification of single monocular color images. Considering that lighting conditions have a significant impact on

weather identification from vehicle-mounted imagery, Pavlic, Rigoll and Ilic (2013) used spectral features and a simple linear classifier to distinguish between clear and foggy weather situations in both day-time and night-time scenarios to improve the visual perception accuracy degradation of in-vehicle cameras in harsh weather and low light conditions. Additionally, CNN-based deep learning algorithms also have been employed (Dhananjaya, Kumar and Yogamani 2021, Triva et al. 2022). From the perspective of hardware, Zhang et al. (2022) mounted visible and infrared cameras in front of

the car to collect day-time and night-time road images. After that, they proposed two single-stream CNN models (visible light and thermal streams) and one dual-stream CNN model developed to classify winter road surface conditions automatically. Samo, Mafeni Mase and Figueredo (2023) argued that a single image includes more than one type of weather. Then, they built a multilabel transport-related dataset of seven weather conditions and assessed different deep-learning models to address multilabel road weather detection tasks. In particular, sensing the transition between these extreme

weather scenes (sunny to rainy, rainy to sunny, and others) is a significant concern for driving safety and is less of a concern. For this, Kondapally et al. (2023) proposed a way to interpolate the intermediate weather transition data using a variational autoencoder and extract its spatial features using VGG. Further, they modelled the temporal distribution of these spatial features using a gated recurrent unit to classify the corresponding transition state. In addition, Aloufi, Alnori and Basuhail (2024) treated weather classification and object detection as a single problem and proposed a new classification network,

which integrated image quality assessment, Super-Resolution Generative Adversarial Network, and a modified version of the YOLO network. This work adds sandy weather recognition, which has yet to be considered in previous research.

However, weather visual data collected by In-vehicle cameras and that of surveillance cameras remain different. Take snow as an example, snow images captured by different visual sensors are presented in Fig.2. Surveillance cameras are usually shot from an overhead view, while the in-vehicle cameras are mainly from a horizontal view. Different shooting angles result

in images with different backgrounds. These efforts take a different perspective than surveillance cameras for weather recognition and provide substantial theoretical and methodological references and guidance for our study.

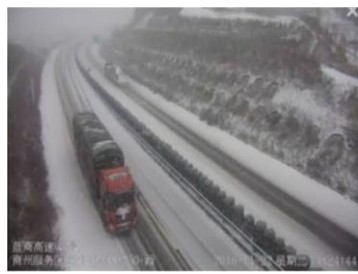 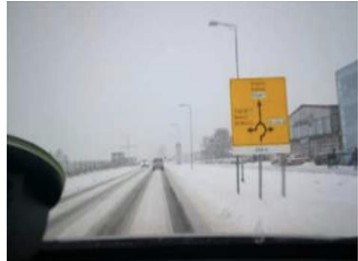 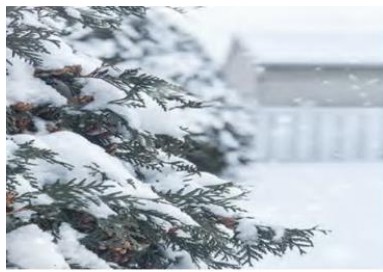

(a) Surveillance camera      (b) In-vehicle camera      (c) Generalized camera

    (Sun et al. 2020)          (Triva et al. 2022)          (Xiao et al. 2021)

**Figure 2: Snow captured by different visual sensors.**



## 2.3. Generalized visual data

Here, the generalized visual data means the pictures/videos taken by visual devices other than surveillance cameras (i.e., cell phones, digital cameras, and web cameras). Nowadays, with rapid dissemination on the Internet and social media platforms, visual data with spatial (geotags) and temporal (timestamps) information can collectively reveal weather information around the world. Based on this, researchers could collect generalized visual data from the Internet or social media platforms for weather condition classification purposes. For example, Chu, Zheng and Ding (2017) used the random forest classifier to build a weather properties estimator; Zhao et al. (2018) propose to treat weather recognition as a multi-label classification task and present a CNN-RNN architecture to identify multi- weather-label from images; Wang, Li and Liu (2018) combine the real-time weather data with the image feature as the final feature vector to identify different weather; Guerra et al. (2018) explored using super-pixel masks as a data augmentation technique, considering different CNN architectures for the feature extraction process when classifying outdoor scenes in a multi-class setting using general-purpose images. Ibrahim, Haworth and Cheng (2019) proposed a new framework named WeatherNet for visibility-related road condition recognition, including weather conditions. WeatherNet takes single-images as input and used multiple deep convolutional neural network (CNN) models to recognise weather conditions such as clear, fog, cloud, rain, and snow. Toğaçar, Ergen and Cömert (2021) used GoogLeNet and VGG-16 models to extract image features and use them as input to construct a spiking neural network for weather classification; Xiao et al. (2021) proposed a novel deep CNN named MeteCNN for weather phenomena classification. Mittal and Sangwan (2023) extracted features using a pre-trained deep CNN model and used transfer learning techniques to build a weather classification framework to save the time and resources needed for the system to work and increase the reliability of the results.

In contrast, as shown in Fig.2, generalized visual data differs significantly from surveillance images/videos in terms of resolution, clarity, background content, etc., and the image characteristics of weather conditions may also differ. Therefore, there are considerable differences in the algorithm design ideas and result accuracy for determining the weather from web images and surveillance images.

Table.1 presents a comparison of surveillance camera-based weather classification/recognition algorithms. Since ordinary surveillance cameras differ in the images captured during daytime and nighttime, the working time is divided into daytime and nighttime (low-light scenarios are categorized as nighttime). Moreover, the weather types that can be recognized/classified are also listed. Combined with previous review and analysis, we can summarise:

**In terms of working time:** Existing studies mainly focus on weather condition in daytime, while that of nighttime is given little attention.

**In terms of weather:** It can be found from Table.1 that the existing works have not yet paid attention to the distinction of graupel, which is more challenging due to its similarity to rain and snow particles.

**In terms of methodology**: mainstream classification methods have shifted from traditional machine learning methods to deep learning methods. For data-driven deep learning methods, a wealthy and high-quality training dataset is the foundation

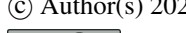



for deep learning model construction. We are pleased to see some datasets for road weather being released (Karaa, Ghazzai
and Sboui 2024, Bharadwaj, Biswas and Ramakrishnan 2016, Guerra et al. 2018). However, existing methods are primarily
focused on single-image information. Compared to images, videos, which contain temporal dependencies between image
frames, are seldom used, although this will help improve the accuracy of recognition results.

**Table 1: Comparison of surveillance camera-based weather classification/recognition studies.**

| | Working time | | Weather can be recognized | | | | | | |
|---|---|---|---|---|---|---|---|---|---|
| | Day time | Nighttime | Rainy | Snowy | Foggy | Cloudy | Sunny | Sandy | Others |
| (Zhao et al. 2011) | √ | | √ | √ | √ | | | | |
| (Shibata et al. 2014) | √ | | √ | √ | | | | | |
| (Li et al. 2014) | √ | | √ | √ | √ | | √ | | |
| (Lee 2017) | √ | | | √ | √ | | | | √ |
| (Dahmane et al. 2018) | √ | √ | √ | √ | √ | | | | |
| (Carrillo and Crowley 2020) | √ | | √ | √ | | | | | |
| (Xia et al. 2020) | √ | | √ | √ | √ | | √ | | |
| (Sun et al. 2020) | √ | | √ | √ | √ | | √ | | |
| (Ramanna et al. 2021) | √ | √ | √ | √ | | | √ | | |
| (Dahmane et al. 2021) | √ | | √ | | √ | | | | |
| (Morris and Yang 2021) | √ | | | √ | | | | | √ |
| (Landry and Akhloufi 2022) | √ | | | √ | | | | | |
| (Khan et al. 2022) | √ | | | √ | | | √ | | |
| (Lü et al. 2023) | √ | | √ | √ | | √ | √ | | |
| Askbom (2023) | √ | | | √ | | | | | √ |
| (Chen et al. 2023) | √ | | √ | √ | √ | √ | √ | | |
| (Wang et al. 2023b) | √ | | | | √ | √ | √ | √ | |

## 3 Methodology

Meteorological and physical studies have explored the size, shape, brightness, and terminal velocity of rain, snow, and
graupel, providing an essential foundation for analyzing their visual characteristics. After analyzing a large number of
surveillance videos, the distinctions between rain, snow, and graupel can be primarily summarized in terms of brightness and





shape from the perspective of video observations. To enhance clarity, we present a comparison of these precipitation types in both visible and near-infrared video footage.

**Brightness:** Ordinary surveillance cameras take visible light video during the day and near-infrared video at night. Therefore, the brightness of the particles differs in the day and night-time images/videos. In the daytime, rain and shrapnel particles have strong forward reflections of visible light and appear brighter than the background. In contrast, snow appears in white; at night, the brightness of the three particles is similar, with little differentiation, as shown in Fig.3.

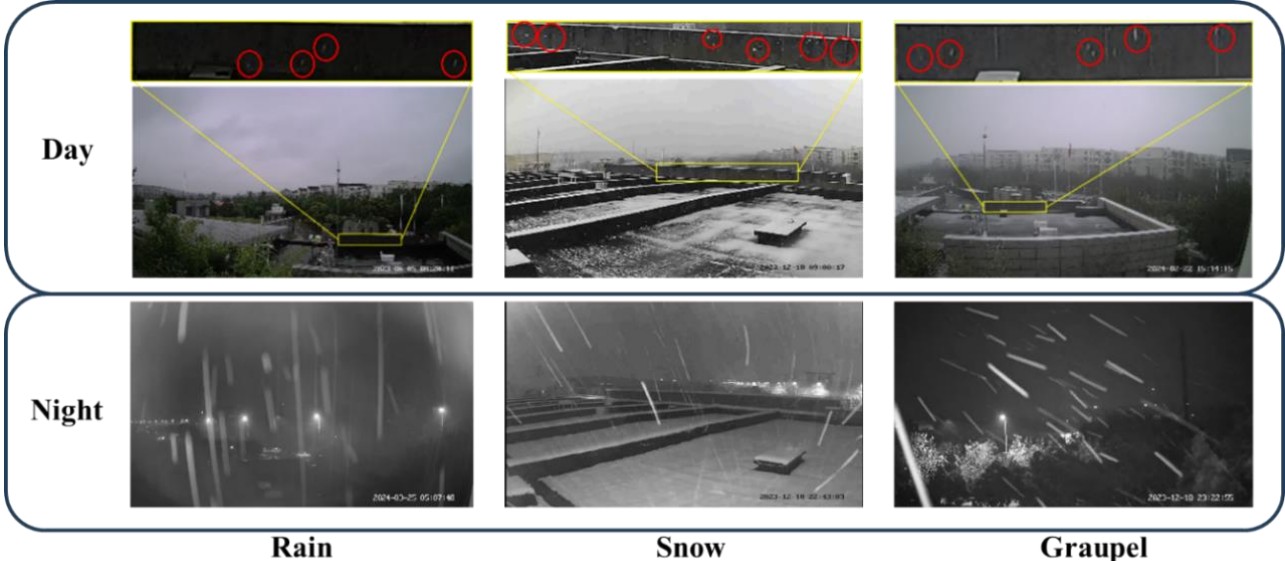

**Figure 3: Comparison of rain, snow, and graupel in day and night surveillance images (Particles are labelled by red circles).**

**Shape**: Due to the long exposure time of surveillance cameras, rain, and graupel particles have a large deformation in surveillance images, usually appearing as lines. These lines describe the trajectories of rain and graupel particles. However, as shown in Fig.3, meteorological studies have pointed out that, in general, the speed of graupel (Heymsfield and Wright

2014, Kajikawa 1975) is greater than that of rain (Montero-Martínez et al. 2009). In combination with the imaging principles of the camera, the trajectory of a graupel particle is longer than that of a rain particle of the same size in the same surveillance camera. Moreover, rain has a greater number concentration value (the number of particles per unit volume) (10 - $10^4$ m$^{-3}$) than graupel (1-10 m$^{-3}$) (Zhang 2016). That is to say, the number of raindrops in the surveillance images is denser compared to graupel. And snow particles have less shape change due to their slower falling speed (≤1 m/s) (Vázquez-Martín,

Kuhn and Eliasson 2021). Overall, the length of rain is wider and shorter than that of graupel, while snow is the shortest.

The analysis shows that distinguishing snow is relatively straightforward, given the significant differences in brightness, shape, and terminal velocity when compared to rain and graupel. However, the primary challenge of this study lies in differentiating between rain and graupel. While there are notable differences in their number concentrations, rain and graupel share similar speeds, shapes, and brightness, which complicates accurate differentiation. Traditional hand-crafted features



fall short in capturing the subtle distinctions between rain and graupel particles, making it necessary to employ deep learning features. Furthermore, accurate classification requires not only spatial features from images but also temporal features from video sequences.

In practical applications, the timeliness of precipitation data extracted from surveillance videos is essential to ensure the value of the data. With the widespread availability of high-definition, full high-definition, and ultra-high-definition

surveillance cameras, video resolution is continuously improving, leading to rapid increases in surveillance data volume. In this context, where numerous cameras generate massive amounts of real-time data, it is vital to consider not only memory and computational resources but also the speed and efficiency of the GHP recognition algorithm, alongside accuracy, for effective processing.

### 3.1. Ground hydrometeor phase classification model construction

In this section, surveillance video-based GHP identification is approached as a video classification task. To balance accuracy, computational speed, and computational load in designing the GHP classification algorithm, a deep learning model that integrates MobileNetV2 and GRU is proposed. First, a pre-trained MobileNetV2 model based on ImageNet is adapted for spatial feature extraction using a transfer learning strategy, enabling it to capture differences between surveillance images from various GHP events. These features are then fed into a GRU network to model the long-term dynamics of the

surveillance video sequence. The structure of the surveillance camera-based GHP identification system is illustrated in Fig. 4.

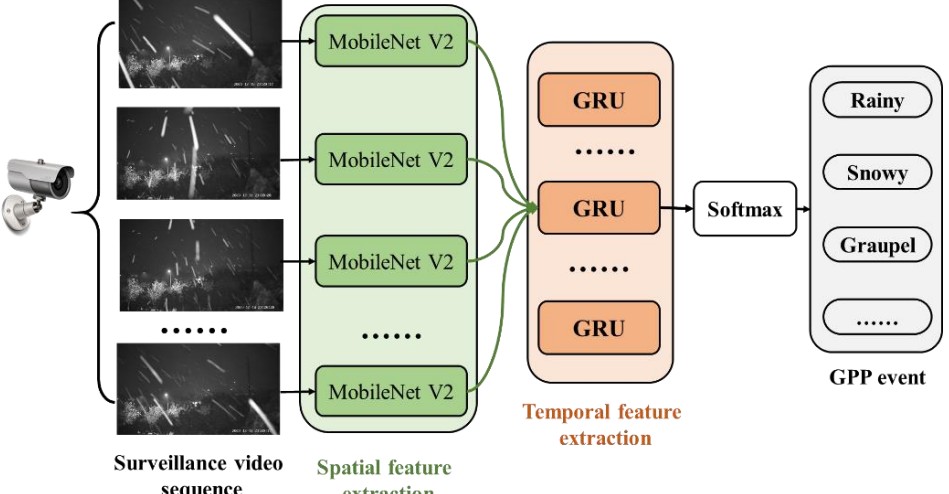

**Figure 4: The structure of the surveillance camera-based ground hydrometeor phase identification system.**

### 3.1.1 Spatial feature extraction model

As analyzed previously, the primary distinctions between rain, snow, and graupel in surveillance images are reflected in their

brightness, shape, and number concentration. The role of the spatial feature extractor is to identify and capture these





differences from surveillance images captured during both daytime and nighttime conditions. MobileNetV2, a lightweight framework, is widely used for visual object classification, recognition, and tracking tasks, offering an effective trade-off between accuracy and model efficiency in terms of size and computational speed (He et al. 2016). These advantages align well with the requirements for spatial feature modeling of various hydrometeor phase particles, making MobileNetV2 the chosen backbone for the spatial feature extraction model. MobileNetV2 builds upon MobileNetV1 and is based on two primary components: the inverted separable convolution (ISC) block and the inverted residual (IR) block.

**ISC block:** This block utilizes a $1 \times 1$ convolution with batch normalization and a ReLU6 activation function ($1 \times 1$ R-Conv) to expand the number of channels in the input feature map. It then calculates the feature maps through depthwise convolution (DW), after which the number of channels is reduced using a linear $1 \times 1$ convolution.

**IR block:** Built upon the ISC block, this block reduces the stride of the DW convolution to 1, maintaining the feature map size before and after processing. It also incorporates a shortcut connection between each residual block, similar to the residual network structure (He et al. 2016). This setup allows the feature maps following the $1 \times 1$ linear convolution ($1 \times 1$ L-Conv) to be added to the input feature maps, completing the calculation of the residual feature maps. The structures of the ISC and IR blocks are shown below.

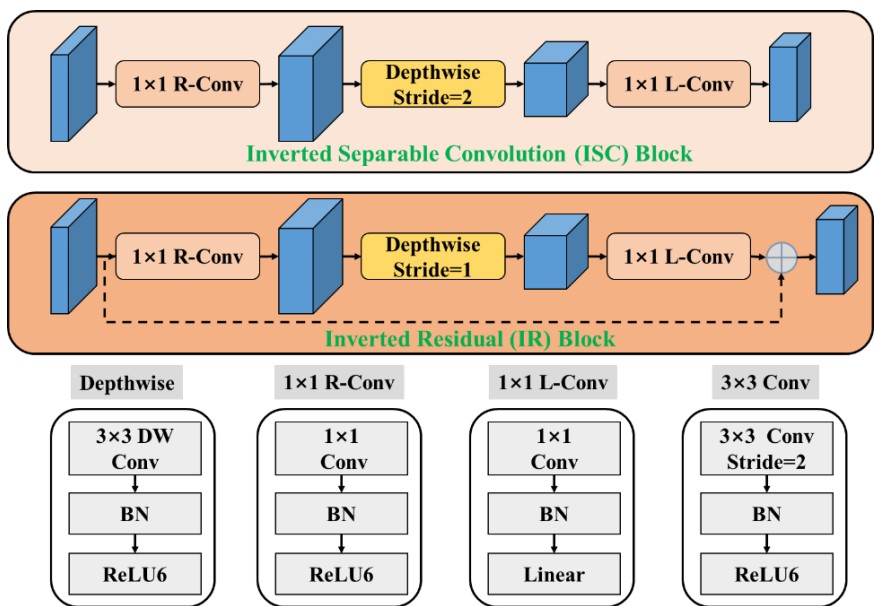

**Figure 5: The structure of the inverted separable convolution (ISC) block and inverted residual (IR) block.**

Figure 6 presents the architecture of the existing MobileNet V2 backbone model. Here, we set the input size of the MobileNet V2 to 512x512x3, and output seven groups of feature maps of different sizes, from $112 \times 112 \times 32$ to $7 \times 7 \times 160$, to the temporal feature extraction model for feature fusion processing.





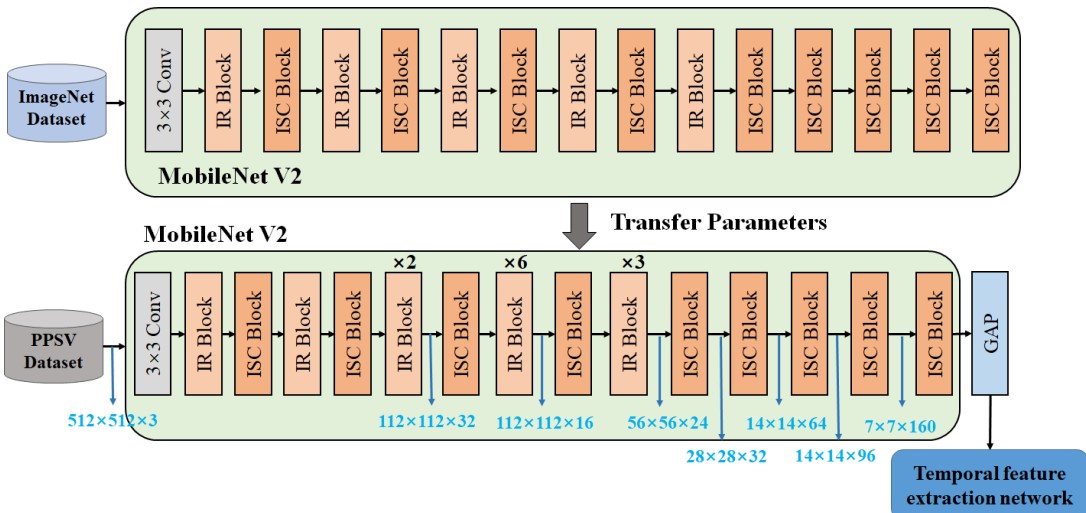


**Figure 6: The structure of MobileNet V2 backbone network.**

Transfer learning involves leveraging knowledge from one task to inform another, eliminating the need for feature extractors to be trained from scratch. This approach accelerates training, reduces the risk of overfitting, and enables the construction of accurate models more efficiently. Previous research has shown that pre-trained models, developed using extensive datasets

like ImageNet, offer an excellent foundation for new tasks where dataset size is limited. Consequently, the final two layers of MobileNetV2 were removed and replaced with global average pooling (GAP), batch normalization (BN), and temporal feature extraction layers (detailed in Section 3), including GRU layers. Finally, the pre-trained MobileNetV2, fine-tuned on the ImageNet dataset, was adapted through transfer learning to extract spatial features of GHP from surveillance images.

### 3.1.2 Temporal feature extraction model

Another critical indicator for differentiating between rain, snow, and graupel particles is their varying falling velocities, as illustrated in Fig.7. While such differences are challenging to detect in single images, they become much more pronounced in video sequences. Consequently, the temporal feature extractor builds upon the spatial feature extractor by capturing the temporal dependencies between adjacent frames, thereby enabling the modeling of falling velocities for rain, snow, and graupel particles. Recurrent Neural Networks (RNN), Long Short-Term Memory (LSTM), and Gated Recurrent Units (GRU)

are widely recognized networks for learning temporal dependencies, effectively leveraging contextual information. This capability is particularly valuable in tasks such as natural language processing, video classification, and speech recognition. RNNs apply recurrent operations to each element in a sequence, where the current computation is influenced by both the current input and previous states. However, traditional RNNs are prone to the vanishing and exploding gradient problems, which limit their effectiveness in capturing long-term dependencies, confining them mostly to short-term dependencies. To

address these limitations, variants such as LSTM and GRU were introduced. These networks are specifically designed to capture long-term dependencies. The GRU, a streamlined version of LSTM, features fewer parameters, making it more



efficient in terms of memory usage and computational speed. A GRU consists of three primary components: the update gate, the reset gate, and the current memory gate.

- **Update gate**: Controls the extent to which previous information is retained and carried forward to future states (Eq. (1)).
- **Reset gate**: Determines the amount of past information that should be discarded (Eq. (2)).
- **Current memory gate**: Computes the current state by integrating the previous hidden state with the current input (Eq. (3)). The final memory is determined as described in Eq. (4).

$$z_t = \sigma(W_z \cdot [h_{t-1}, x_t] + b_z) \tag{1}$$

$$r_t = \sigma(W_r \cdot [h_{t-1}, x_t] + b_r) \tag{2}$$

$$\tilde{h}_t = \tanh(W \cdot [r_t * h_{t-1}, x_t] + b) \tag{3}$$

$$h_t = (1 - z_t) * h_{t-1} + z_t * \tilde{h}_t \tag{4}$$

Where, $W_z$, $W_r$, and $W$ are learnable weight matrices, $h_{t-1}$ is the previous hidden state, $x_t$ is the input vector, $\sigma$ and tanh are the sigmoid and tanh activation function, $*$ represents the Hadamard product, and $b_z$, $b_r$ and $b$ are biases.

After extracting spatial features, they are input into a GRU layer with 93 hidden units to capture temporal dependencies. The outputs from the GRU layer are concatenated and passed through a dense layer. Following the dense layer, a batch normalization layer is applied, which is subsequently connected to a fully connected output layer. This final layer uses a SoftMax activation function to classify the GHP event from the surveillance video sequence. The spatial feature map has an input shape of $1280 \times N$, where N represents the length of the video sequence utilized for temporal dependency modeling. The impact of different values of N on the accuracy of GHP classification is analyzed in Section 4.4.

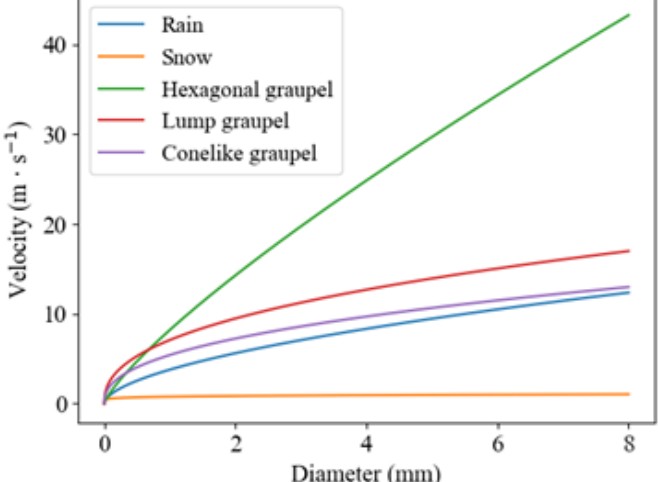

**Figure 7: Comparison of terminal velocities of different particles.**





## 3.2. Dataset building

For training and testing the deep-learning model, a new Hydrometeor Surveillance Video (HSV) dataset was constructed. As
illustrated in Fig. 8, 20 surveillance cameras were deployed at the National Benchmark Climate Station in Nanjing, Jiangsu,
China. Of these, videos from 17 cameras (IDs 4 to 20) were utilized to build the dataset, while the remaining 3 cameras (IDs
1 to 3) were reserved for real-world precipitation observation experiments, as discussed in Section 4.5.

Considering the broad range of potential applications, the deployed urban surveillance cameras exhibit substantial variation
in parameters, including resolution (960×720, 1280×960, 1920×1080, 2592×1944), focal length (4mm, 6mm, 8mm, 12mm),
and frame rate (15fps, 20fps, 25fps). This diversity ensures that the collected video data reflects real-world surveillance
conditions, thereby minimizing the gap between the performance of the deep-learning model on the HSV dataset and its
applicability in real-world scenarios.

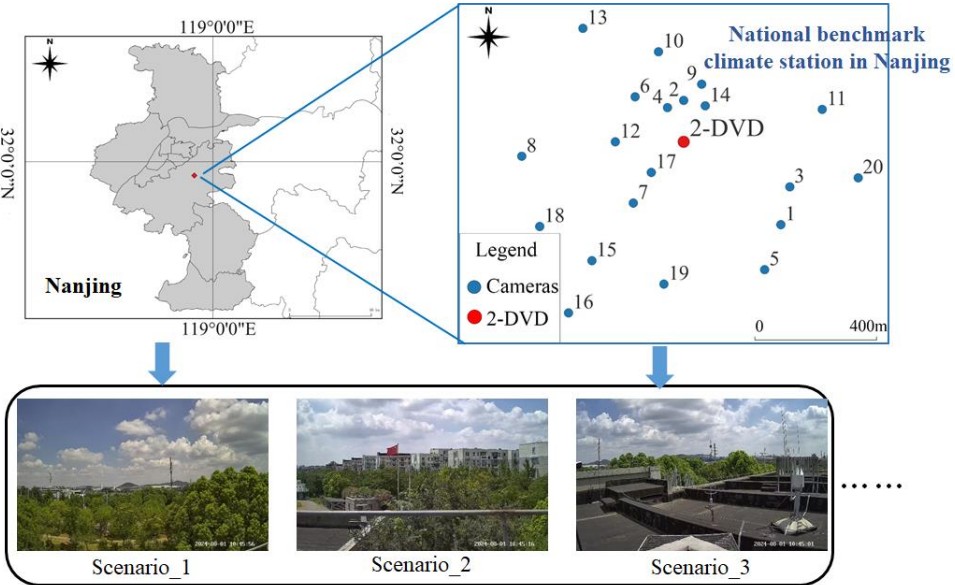

**Figure 8: Overview of the study area.**

After a long period of observation (starting from March 2023 and ending in July 2024), we captured a huge amount of
surveillance video data for different hydrometeor phases. During this period, we captured extreme precipitation surveillance
videos such as snowstorms with intensities of 27 mm/h and heavy rainfalls of 195 mm/h. These rare and precious
precipitation scenarios play an important role in improving the generalization and diversity of our dataset. Finally, about 94
hours of surveillance videos from day to night were selected and categorized into four categories: rainfall, snowfall, graupel,
and no precipitation. The videos were divided into segments, each of which was 5 seconds in length, with a frame rate
ranging from 15-25 fps, and were saved in .mp4 format. More details can be found in Table.2.

In addition, the Two-Dimensional Video Disdrometer (2-DVD), a professional precipitation measurement instrument, works
in synchronization with the surveillance cameras to provide the true value/label of the hydrometeor phase for the
surveillance videos. Simultaneous observations by researchers are also conducted, and their observed data are used to refine





the 2DVD measurements, ensuring the accuracy of the true value/label of GHP. The maximum distance between the 2DVD and the camera is 1 km, which ensures that the hydrometeor phase observed by the two is the same.

**Table 2: Description of the HSV dataset.**

|  | Day-time | Night-time | Note |
|---|---|---|---|
| No precipitation | 6903 | 7378 |  |
| Rainy | 11183 | 11249 | Precipitation intensity: 0 ~195 mm/h |
| Snowy | 8086 | 7938 | Precipitation intensity: 0 ~ 27 mm/h |
| Graupel | 7272 | 7692 | Precipitation intensity: 0 ~ 23 mm/h |

## 4. Experiment and discussion

### 4.1. Experimental environment


Our experiments were performed on a workstation with Ubuntu 11.2.0 (Linux 5.15.0-25-generic) for the operating system. More specifications are as follows:

- 4× Intel Xeon Silver 4216 CPU@2.10 GHz (32 cores);
- 8×NVIDIA GEFORCE GTX2080Ti graphics cards equipped with 11 GB GDDR6 memory;
- 188 GB RAM;
- Python 3.9.16;
- TensorFlow 2.4.1, Scikit-learn 1.2.1, and Keras 2.4.3 libraries;
- CUDA 11.8 and CUDNN 8;

### 4.2. Evaluation metrics

To evaluate the GHP classifiers, we selected 3 different established metrics: the balance accuracy, weighted precision, and weighted recall. The balance accuracy metric ( *Accuracy* ) is described as:

$$Accuracy = \frac{1}{n}\sum_{i=1}^{n}\frac{TP_i}{S_i} \tag{5}$$

Where $TP_i$ and $S_i$ stand for the number of True Positive and sample size of class $i$, respectively.

The weighted precision metric ( *Precision* ) can be calculated as follows:

$$Precision = \sum_{i=1}^{n}\frac{TP_i}{\left(TP_i + FP_i\right)} * r_i \tag{6}$$





Where $FP_i$ is the number of False Positive of class $i$ and $r_i$ is the ratio between the number of samples of class $i$ and the total number of samples.

The weighted recall ( *Recall* ) is calculated as follows:

$$Recall = \sum_{i=1}^{n} \frac{TP_i}{\left(TP_i + FN_i\right)} * r_i \tag{7}$$

### 4.3. Model training details

The HSV dataset was split into training, validation, and test datasets according to the ratio of 7:2:1. Training and validation sets were employed to construct the deep learning model. To analyze the performance of the proposed method, some classical CNN models with ImageNet pre-trained weights such as: DenseNet 121 (Huang et al. 2017), EfficientNet B0 (Tan and Le 2019), Inception V3 (Szegedy et al. 2016), and ResNet 50 (He et al. 2016) are used to extract the spatial features of precipitation images, while some commonly used neural networks for temporal signal analysis like RNN, LSTM, 1D-CNN (Kiranyaz et al. 2021), and Bi-LSTM (Huang, Xu and Yu 2015) are employed to extract the temporal features of precipitation surveillance videos, respectively. In terms of realization, transfer learning is exploited, and the last layer of each spatial extraction architecture (i.e., the fully connected (FC) and Softmax layers) was deleted and replaced with two layers: GAP and BN, to extract the deep spatial features based on transfer learning. Second, the spatial features were sent to temporal feature extraction networks listed above. The Softmax function was used as a classifier to identify GHP. Thus, a total of 25 deep-learning algorithms are constructed and compared. The hyper-parameters deep-learning models were set as follows:

**Table 3: Hyper-parameters for deep-learning models.**

| Spatial feature extractor | | Temporal feature extractor | |
|---|---|---|---|
| Hyper-parameters | Value | Hyper-parameters | Value |
| Learning Rate | 0.001 | Learning Rate | 0.001 |
| Batch Size | 32 | Batch Size | 32 |
| Number of Epoches | 100 | Number of Epoches | 100 |
| Dropout Rate | 0.5 | Dropout Rate | 0.5 |
| Filters | [64, 128, 256, 512] | Activation Functions | Sigmoid |
| Activation Functions | ReLU | Weight Initialization | Glorot Initialization |
| Weight Initialization | He Initialization | Loss Function | Mean Squared Error |





### 4.4. Experiments on the HSV dataset

Given the length of the input frames on the temporal feature extraction, we evaluated the performance of different algorithms when screening 5, 10, and 15 frames per second from the video clips for comparison. The results of different numbers of frames per second (NFS) as input to temporal feature extraction are shown in Tables 4, 5, and 6.

**Table 4: Comparison of GHP recognition by different deep-learning algorithms (NFS = 5).**

| | | RNN | LSTM | GRU | 1D-CNN | Bi-LSTM |
|---|---|---|---|---|---|---|
| DenseNet 121 | *Accuracy* | 0.9479 | 0.8856 | **0.9656** | 0.9295 | 0.9334 |
| | *Precision* | 0.9428 | 0.8874 | **0.9639** | 0.9278 | 0.9335 |
| | *Recall* | 0.9415 | 0.8903 | **0.9617** | 0.9322 | 0.9341 |
| EfficientNet B0 | *Accuracy* | 0.9051 | 0.9508 | 0.9253 | 0.9550 | 0.9558 |
| | *Precision* | 0.8993 | 0.9522 | 0.9245 | 0.9408 | 0.9574 |
| | *Recall* | 0.9072 | 0.9498 | 0.9208 | 0.9558 | 0.9497 |
| Inception V3 | *Accuracy* | 0.8887 | 0.9448 | 0.9237 | 0.9418 | 0.9442 |
| | *Precision* | 0.8693 | 0.9414 | 0.9201 | 0.9373 | 0.9485 |
| | *Recall* | 0.8921 | 0.9431 | 0.9199 | 0.9487 | 0.9471 |
| ResNet 50 | *Accuracy* | 0.9390 | **0.9660** | 0.9581 | 0.9287 | 0.9558 |
| | *Precision* | 0.9302 | **0.9599** | 0.9517 | 0.9189 | 0.9576 |
| | *Recall* | 0.9323 | **0.9674** | 0.9585 | 0.9207 | 0.9511 |
| MobileNet V2 | *Accuracy* | 0.9576 | 0.9496 | **0.9671** | **0.9610** | 0.9440 |
| | *Precision* | 0.9526 | 0.9469 | **0.9597** | **0.9624** | 0.9487 |
| | *Recall* | 0.9513 | 0.9502 | **0.9634** | **0.9626** | 0.9398 |

**Table 5: Comparison of GHP recognition by different deep-learning algorithms (NFS = 10).**

| | | RNN | LSTM | GRU | 1D-CNN | Bi-LSTM |
|---|---|---|---|---|---|---|
| DenseNet 121 | *Accuracy* | 0.9104 | 0.9376 | 0.9507 | 0.9483 | **0.9620** |
| | *Precision* | 0.9115 | 0.9336 | 0.9487 | 0.9402 | **0.9643** |
| | *Recall* | 0.9127 | 0.9258 | 0.9512 | 0.9477 | **0.9601** |
| EfficientNet B0 | *Accuracy* | 0.9247 | 0.9274 | 0.9368 | 0.9516 | 0.9335 |
| | *Precision* | 0.9202 | 0.9243 | 0.9402 | 0.9475 | 0.9278 |
| | *Recall* | 0.9253 | 0.9148 | 0.9335 | 0.9514 | 0.9238 |
| Inception V3 | *Accuracy* | 0.8962 | 0.9336 | 0.9498 | 0.9476 | **0.9641** |
| | *Precision* | 0.9012 | 0.9345 | 0.9464 | 0.9427 | **0.9578** |
| | *Recall* | 0.8913 | 0.9402 | 0.9352 | 0.9453 | **0.9622** |
| ResNet 50 | *Accuracy* | 0.9152 | 0.9306 | **0.9651** | 0.9526 | 0.9521 |
| | *Precision* | 0.9158 | 0.9267 | **0.9645** | 0.9516 | 0.9502 |
| | *Recall* | 0.9127 | 0.9317 | **0.9661** | 0.9548 | 0.9544 |
| MobileNet V2 | *Accuracy* | 0.9108 | 0.9211 | **0.9677** | 0.9433 | 0.9546 |
| | *Precision* | 0.9121 | 0.9217 | **0.9644** | 0.9423 | 0.9549 |
| | *Recall* | 0.9098 | 0.9159 | **0.9758** | 0.9409 | 0.9601 |





**Table 6: Comparison of GHP recognition by different deep-learning algorithms (NFS = 15).**

|  |  | RNN | LSTM | GRU | 1D-CNN | Bi-LSTM |
|---|---|---|---|---|---|---|
| DenseNet 121 | *Accuracy* | 0.9291 | 0.8821 | **0.9490** | 0.9067 | 0.9265 |
|  | *Precision* | 0.9322 | 0.8689 | **0.9347** | 0.8874 | 0.9202 |
|  | *Recall* | 0.9107 | 0.8778 | **0.9426** | 0.8955 | 0.9178 |
| EfficientNet B0 | *Accuracy* | 0.8857 | 0.9105 | 0.9320 | 0.9012 | 0.9272 |
|  | *Precision* | 0.8656 | 0.9047 | 0.9189 | 0.9036 | 0.9178 |
|  | *Recall* | 0.8645 | 0.8993 | 0.9275 | 0.8998 | 0.9302 |
| Inception V3 | *Accuracy* | 0.9124 | 0.9271 | 0.9225 | 0.9254 | 0.9404 |
|  | *Precision* | 0.9111 | 0.9215 | 0.9057 | 0.9114 | 0.9287 |
|  | *Recall* | 0.9074 | 0.9303 | 0.9154 | 0.9233 | 0.9444 |
| ResNet 50 | *Accuracy* | 0.9087 | 0.8952 | 0.9385 | **0.9554** | 0.9248 |
|  | *Precision* | 0.8954 | 0.9012 | 0.9346 | **0.9547** | 0.9301 |
|  | *Recall* | 0.9111 | 0.8872 | 0.9245 | **0.9504** | 0.9287 |
| MobileNet V2 | *Accuracy* | 0.8938 | 0.9275 | **0.9577** | 0.9189 | 0.9350 |
|  | *Precision* | 0.8952 | 0.9245 | **0.9542** | 0.9105 | 0.9374 |
|  | *Recall* | 0.8911 | 0.9147 | **0.9553** | 0.9233 | 0.9326 |


The results indicate that when the number of frames (NFS) is set to 5, the accuracy of our proposed method closely aligns with those of the DenseNet 121+GRU, ResNet 50+LSTM, and MobileNet V2+1D-CNN models. When the NFS is increased to 10, the performance of our proposed method, DenseNet 121+Bi-LSTM, Inception V3+Bi-LSTM, and ResNet 50+GRU converges, showing minimal differences in classification accuracy, which ranges from 0.960 to 0.967. However, with NFS at

15, the accuracy of our proposed method surpasses that of DenseNet 121+GRU and ResNet 50+1D-CNN models, though it slightly declines to approximately 0.949 to 0.957.

The observed improvement in deep learning model performance when increasing NFS from 5 to 10 frames can be attributed to the enriched temporal features provided by the additional frames. These features enhance the models' ability to differentiate between various hydrometeor phases (GHP). Nonetheless, when the NFS reaches 15 (equivalent to 75 frames

per video clip), the lengthier temporal sequences challenge the RNN and 1D-CNN architectures, resulting in reduced classification accuracy. In contrast, the GRU architecture, with its more compact structure and computational efficiency, facilitates faster aggregation during training on the HSV dataset, allowing it to sustain high accuracy even with longer NFS. Our proposed algorithm demonstrates classification accuracies of 0.9671, 0.9677, and 0.9577 across the three experimental settings, thereby exhibiting consistently superior stability compared to other methods.

To further assess model performance, confusion matrices are utilized as visual tools, elucidating the relationship between actual and predicted classifications. Figure 9 presents the confusion matrices for our proposed deep learning models on the HSV dataset, detailing hydrometeor phase discrimination capabilities (confusion matrices for comparison models are provided in Appendix A). In these matrices, columns denote true labels, while rows represent predicted classifications by different algorithms. Additionally, a violin plot in Fig.10 quantifies the GHP classification performance across models,

providing further insight into the comparative strengths of each approach.





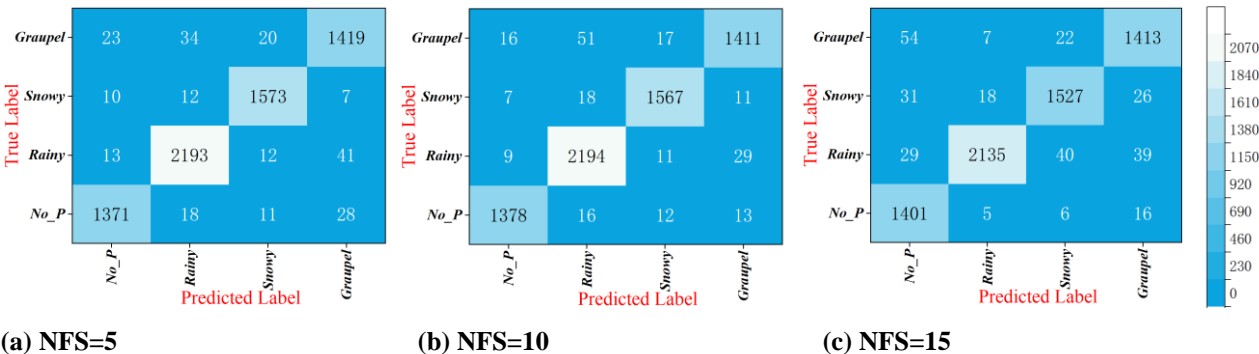

**(a) NFS=5**         **(b) NFS=10**         **(c) NFS=15**

**Figure 9: Confusion matrix of proposed algorithm (No_P means No precipitation).**

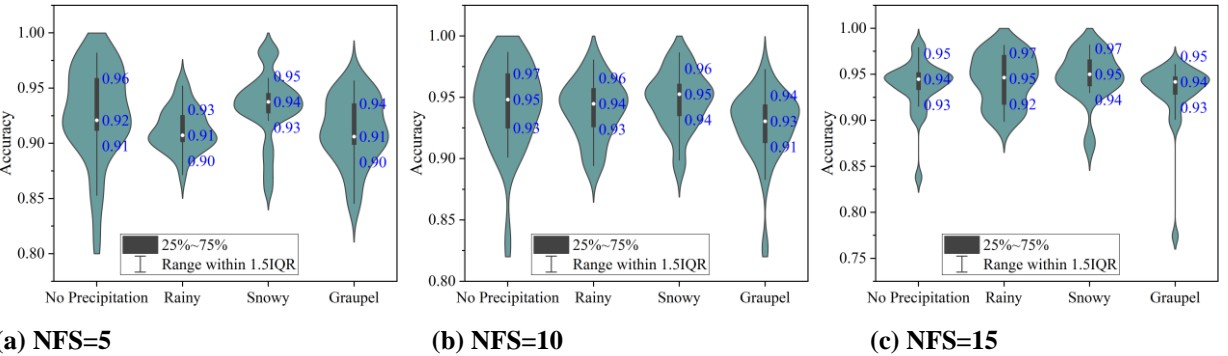

**(a) NFS=5**         **(b) NFS=10**         **(c) NFS=15**

**Figure 10: Violin plot of deep-learning algorithms' performance for GHP classification.**

Overall, the classification accuracy of above listed algorithms is slightly higher for "no precipitation" and "snowy" conditions compared to "rainy" and "graupel." The confusion matrices indicate that the primary source of misclassification among the algorithms lies in differentiating between "rainy" and "graupel" events. This issue arises due to the distinct shape, color, and falling velocity of snow particles, which starkly contrasts with rain and graupel, thereby making snowy conditions easier to classify. As discussed in Section 3, both "rainy" and "graupel" share similar visual and temporal characteristics in both daytime visible and nighttime near-infrared videos, posing significant challenges for the classification algorithms.

Furthermore, Fig.9 illustrates occasional misclassification between "no precipitation" and other GHPs. Upon analysing the HSV dataset, it was observed that these errors typically occurred during low-intensity precipitation events, where only a minimal number of rain, snow, or graupel particles were present. While the 2-DVD device—known for its high sensitivity—can detect such subtle precipitation events, capturing these minute particles in surveillance videos remains challenging, particularly when affected by lighting conditions and external environmental factors within the camera's field of view.

Our proposed method effectively balances temporal and spatial features in precipitation surveillance videos, achieving classification accuracies for "no precipitation," "rainy," "snowy," and "graupel" of 0.9454, 0.9652, 0.9657, and 0.9439, respectively, at NFS = 5. The accuracies improve to 0.9713, 0.9795, 0.9775, and 0.9438 at NFS = 10, and 0.9811, 0.9519, 0.9532, and 0.9445 at NFS = 15. These results demonstrate our algorithm's consistently high and balanced accuracy across





all hydrometeor phase types, with NFS = 10 being the optimal setting. This configuration has been adopted for real-world precipitation observation experiments, as detailed in Section 4.5.

### 4.5. Real-world experiments

Next, we evaluate the performance of the proposed method in real precipitation scenarios. The 2-DVD measurements, calibrated through simultaneous observations by researchers, are used as the true value to validate the effectiveness of the verification method. Here, six precipitation events are selected, including three types of precipitation scenarios: rain, snow, and shrapnel from day-time to night-time, and the duration of each precipitation event is 2 hours. More details about each precipitation event are presented in Table.7. Considering the impact of wind on the trajectory and falling speed of

precipitation particles, we have taken further measures to enhance the robustness and accuracy of the model. Specifically, we installed an anemometer next to the surveillance camera to capture real-time changes in wind speed and direction. This measure provides the model with relevant wind field data to better account for wind interference when predicting GHP. Since the orientation of the surveillance camera is generally fixed, the wind direction data collected by the anemometer can be combined with the camera's viewpoint to calculate the relative orientation of the wind to the camera lens.

**Table 7: Precipitation duration of real-world experiments.**

|  |  | Date (UTC+8) |
|---|---|---|
| Rainy | Day-time | July 12, 2024, 8.00-10.00 a.m. |
|  | Night-time | July 19, 2023, 1.30-3.30 a.m. |
| Snowy | Day-time | December 18, 2023, 9.00-11.00 a.m. |
|  | Night-time | December 18, 2023, 9.30-11.30 p.m. |
| Graupel | Day-time | February 22, 2024, 12.30-3.30 p.m. |
|  | Night-time | December 15, 2023, 10.00-12.00 p.m. |

Three surveillance cameras (ID: 1, 2, and 3, as shown in Fig.8) are used for real-world precipitation observation experiments. The key parameters of these three EZVIZ C5 series cameras are as following:

- image resolution: 2592×1944, 1920×1080, 1280×960;
- focal lengths: 4 mm, 6 mm, and 8 mm;
- frame rate: 15fps, 20fps, 25fps;

The three selected surveillance cameras simultaneously recorded the same precipitation event, although each camera captured a different scene. This arrangement supports the stability analysis of the deep learning algorithm. The field of view of the three cameras is shown in Fig.11.





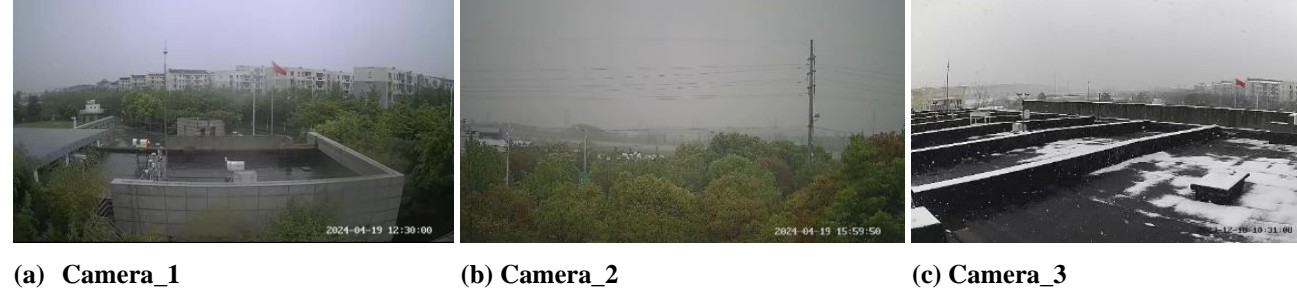


| (a) Camera_1 | (b) Camera_2 | (c) Camera_3 |

**Figure 11: The field of view of the three cameras.**

In line with the findings presented in Section 4.4, surveillance videos were processed by the model with a 5-second interval

between frames, capturing 10 frames per second (NFS=10) for temporal feature extraction. The classification performance

for various GHPs from Camera_1 is illustrated in Fig.12, while the results from Camera_2 and Camera_3 are provided in

Appendix B. In practical applications, the occurrence of false alarms significantly reduces the effectiveness of GHP

recognition system. Therefore, the "no precipitation" label was also included in the evaluation. More detailed experimental

results can be found in Table 8.

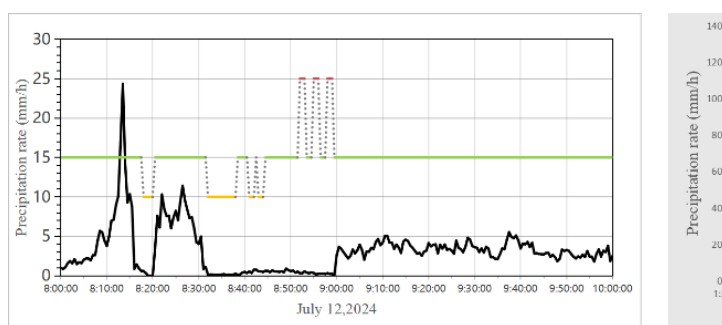

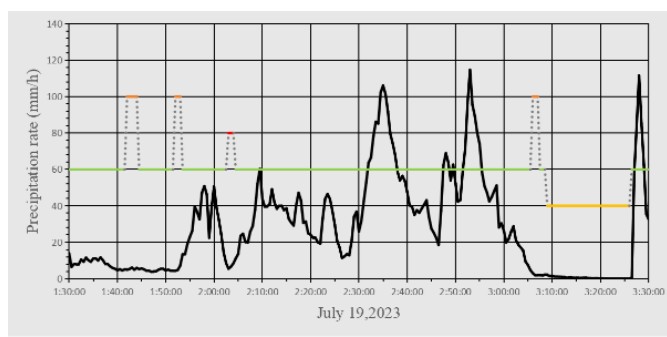

**(a) Day-time rainy**                      **(b) Night-time rainy**

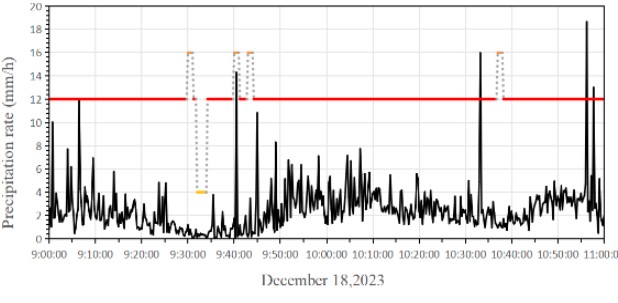

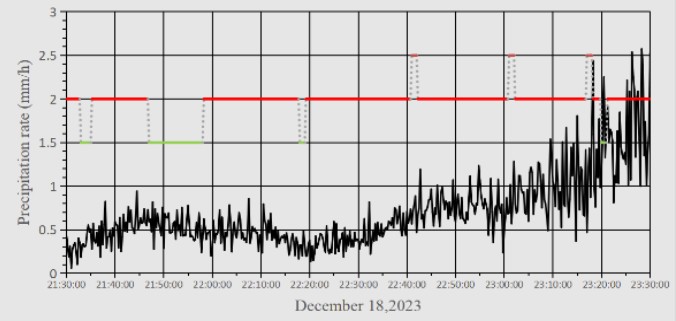

**(c) Day-time snowy**                          **(d) Night-time snowy**





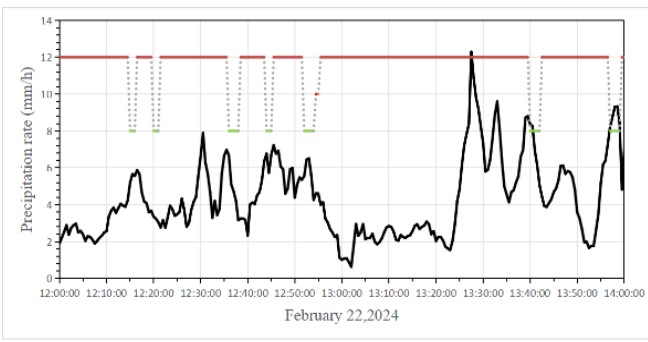
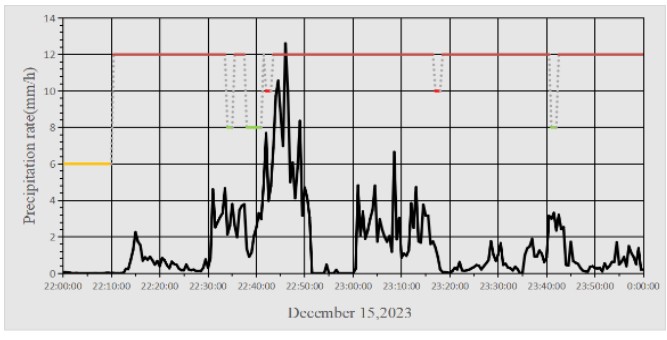

**(e) Day-time graupel**          **(f) Night-time graupel**

**Figure 12: Real-world GHP classification in camera_1.**

( ——— : rainy; ——— : snowy; ——— :graupel; ——— : no precipitation; The black curve represents the precipitation intensity readings from the 2-DVD)

**Table 8: Performance of the proposed method for real-world hydrometeor phase identification.**

|  |  | Camera_1 | Camera_2 | Camera_3 | Average |
|---|---|---|---|---|---|
| Rainy | Day-time | 0.9252 | 0.9174 | 0.9086 | 0.9171 |
|  | Night-time | 0.9151 | 0.9134 | 0.9052 | 0.9112 |
| Snowy | Day-time | 0.9673 | 0.9657 | 0.9581 | 0.9637 |
|  | Night-time | 0.9534 | 0.9606 | 0.9625 | 0.9588 |
| Graupel | Day-time | 0.8756 | 0.8675 | 0.8923 | 0.8785 |
|  | Night-time | 0.8581 | 0.8583 | 0.8836 | 0.8667 |
| No Precipitation | Day-time | 0.9874 | 0.9781 | 0.9578 | 0.9744 |
|  | Night-time | 0.9785 | 0.9672 | 0.9657 | 0.9705 |
| Average |  | 0.9326 | 0.9285 | 0.9292 | 0.9301 |

In real-world precipitation observation experiments, the proposed method achieves an average classification accuracy of 0.9301. Specifically, under daytime conditions, the method achieves classification accuracies of 0.9171, 0.9670, 0.8785, and 0.9744 for "no precipitation," "rainy," "snowy," and "graupel," respectively. Under nighttime conditions, the corresponding classification accuracies are 0.9112, 0.9225, 0.8667, and 0.9705. Overall, the classification accuracies for "rainy" and "graupel" are comparatively lower, which aligns with the results observed in the HSV dataset. Upon further analysis, many misclassifications occur when precipitation intensity is low, often being misidentified as "no precipitation". This is likely due to the fact that under low precipitation intensity, the number concentration of the precipitation particles is small, making them difficult to detect in both daytime and nighttime videos. This issue is also supported by the data shown in Fig.12 and



Appendix B, where low-intensity precipitation events are hard to identify. As analysed in Section 4.4, the validation data used in this study mainly comes from the 2-DVD, which captures falling precipitation particles using a linear array scanning method, offering a high sensitivity to precipitation particles (Kruger and Krajewski 2002). This allows the 2-DVD to detect precipitation events even at low particle concentrations, while such events may be missed in the surveillance video due to frame rate limitations or lighting conditions. This discrepancy leads to misclassification of some precipitation events as "no precipitation". Moreover, as our method employs frame skipping when feeding video frames into the deep learning network (NFS=10), the probability of missing precipitation particles is further increased. Using more video frames as inputs to the deep learning model is a feasible approach. This would increase the temporal capture of precipitation particles, thereby reducing the likelihood of misclassification. However, this approach would also lead to an increase in model complexity and computational delay. Therefore, while optimizing the input data strategy, a balance needs to be found between model accuracy and computational efficiency to ensure real-time performance and stability in practical applications.

As analysis in Section 3, it is evident that the visual characteristics of rain, snow, and graupel differ between daytime and nighttime surveillance videos. Especially, during nighttime, present additional challenges due to varying light sources such as streetlights, vehicle headlights, and other ambient lighting interferences, which can significantly impact image features of GHP. Despite these variations, the results indicate that the proposed algorithm performs consistently well under both day and night conditions. This robustness underscores the algorithm's ability to effectively capture and distinguish the spatiotemporal features of various GHPs across complex illumination scenarios, making it highly suitable for GHP recognition tasks throughout the entire day. Furthermore, these findings suggest that exploring the distinctions between different GHPs from both spatial and temporal dimensions provides a reliable benchmark for future model enhancements. This also lays a solid foundation for refining deep learning models to improve their generalization ability across diverse real-world surveillance conditions.

Based on the previous introduction to camera parameters, the three surveillance cameras used in this study have different fields of view, image resolutions, and frame rates. In particular, differences in frame rates often imply variations in exposure time, leading to discrepancies in the captured images of the same precipitation particle, such as rainstreak length and width. This increases the challenge of distinguishing GPH. Despite the significant differences in camera parameters, our algorithm demonstrates consistent performance across all devices, exhibiting remarkable stability. This robustness can be attributed to two key factors. First, the self-constructed HSV dataset is derived from a large number of real-world urban surveillance scenarios captured by cameras with varying parameters. Second, the results indicate strong generalization capability, thereby enhancing stability in cross-camera observations. Our method effectively integrates the temporal and spatial (image) characteristics of different hydrometeors, combining spatiotemporal-based features in a unified framework. This resilience to variations in camera parameters makes our approach particularly suitable for large-scale practical applications, as urban surveillance cameras typically exhibit substantial configuration differences. The demonstrated robustness ensures reliable performance across diverse surveillance environments, enhancing the applicability of our proposed algorithm in real-world scenarios.





The influence of wind on precipitation particles mainly lies in altering their movement direction and falling speed, which, in turn, affects their representation in the camera's field of view. To investigate this, we examined the impact of wind speed and direction on the performance of the precipitation phase recognition method. As shown in Fig.12, surveillance cameras
capture 2D images to represent a 3D space, meaning that wind causes particles to enter the camera's field of view from the front, rear, left, or right. The left and right directions are symmetrical, so particles entering from these two directions exhibit similar characteristics in the image or video. This is because wind from the left or right does not significantly affect the projected shape of the particles. Furthermore, since the directions are symmetrical, the paths, speeds, and variations of particles entering from these directions present similar features in the image. In contrast, particles entering from the front and
rear exhibit different visual effects when projected onto the image plane. This difference is due to variations in the camera's focal length, field depth, and distance. Particles entering from the front appear to grow larger in the video, whereas those from the rear appear smaller. This phenomenon arises from the pinhole imaging principle followed by the monitoring camera, where objects closer to the camera appear larger in the field of view, and those farther away appear smaller. Thus, particles from the front and rear present distinct characteristics in the image, following the "near large, far small" imaging rule. As
shown in Fig.13, we categorized particle's relative direction to surveillance cameras (orientation of the surveillance camera as $0°$) into four classes:

- Normal: Particles fall vertically and enter the camera's field of view, showing typical image/video features.
- Side direction: Particles enter from the left or right, presenting similar characteristics due to symmetry.
- Front direction: Particles enter from the front, with their size changing according to the distance from the camera.
- Rear direction: Particles enter from the rear, with their size decreasing as they move farther away.

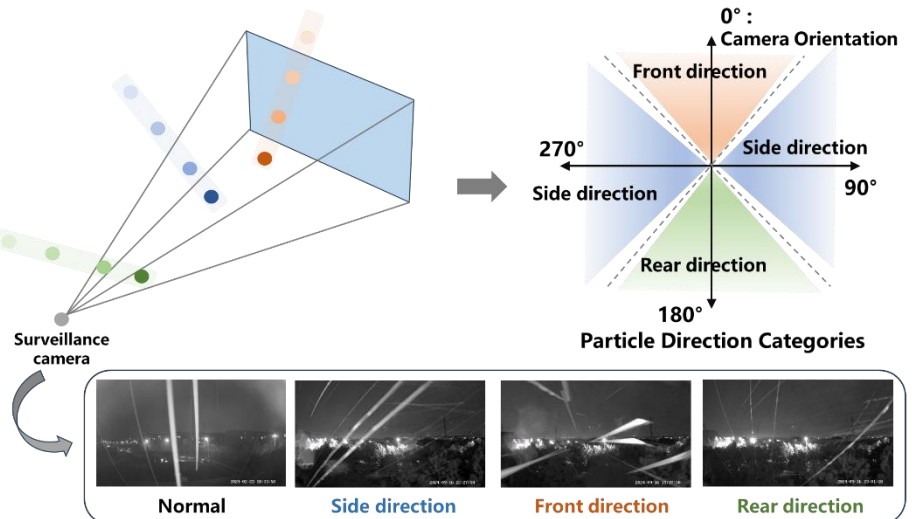

**Figure 13: The definition of particle's relative direction to surveillance cameras.**

Since the orientations of the three surveillance cameras are known, the relative wind direction of camera can be calculated by:

$$\triangle\theta = (\theta_w - \theta_c) \bmod 360 \tag{8}$$





Where, $\theta_w$ represents the wind direction provided by the anemometer, $\theta_c$ represents the orientation of the surveillance camera.

As shown in Fig.13, the particle's relative direction to the camera can be determined based on the value of $\triangle\theta$, as follows: Side direction: $\triangle\theta \in [45, 135) \bigcup [225,325)$; Front direction: $\triangle\theta \in [325,45)$; Rear direction: $\triangle\theta \in [135,225)$. Next, we have statistically analyzed the recognition accuracy of GHPs under different wind speeds and directions. The data in the figure

represent the average values from the three surveillance cameras. Considering that wind speeds ranging from 0 to 1 m/s cause minimal tilting of the precipitation particles, these cases are classified as "Normal" and are not separately reported.

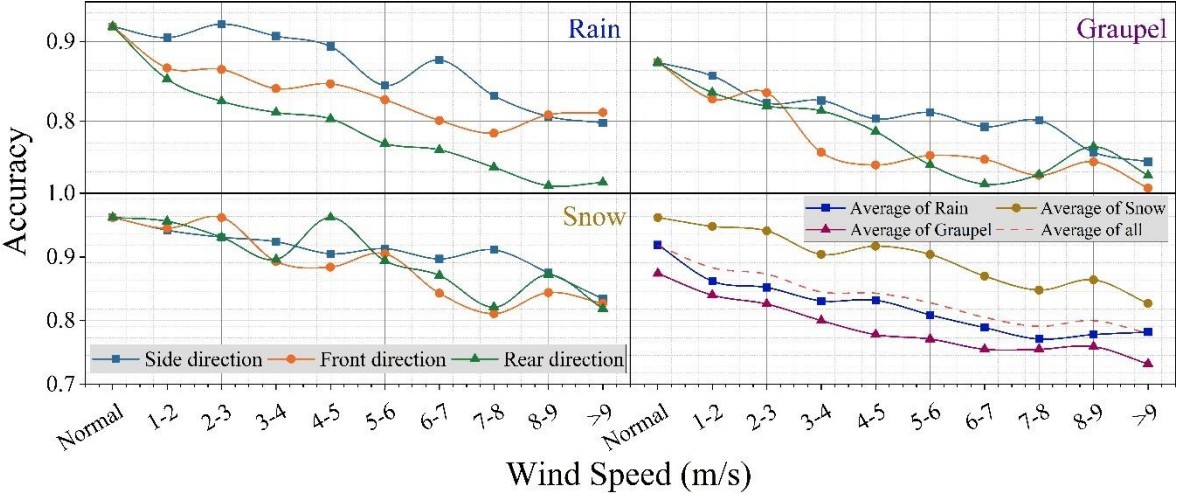

**Figure 14: The influence of wind speed and direction on the accuracy of GHP classification.**

Overall, wind negatively impacts the classification accuracy of GHP, with its influence becoming more pronounced as wind
speed increases. In particular, distinguishing between rain and graupel presents greater challenges under windy conditions, as the model's classification accuracy deteriorates significantly with increasing wind speed. Nevertheless, when wind speed is below 5 m/s, the proposed method still achieves an accuracy of approximately 0.8 for rain-graupel classification, indicating that the method remains effective within this wind speed range. In comparison, when wind speed is below 6 m/s, the proposed model maintains a classification accuracy above 0.9 for snow under different wind directions, demonstrating high
reliability. Furthermore, the influence of particle direction (i.e., wind direction) is also significant and follows certain patterns. For instance, the classification accuracy for particles arriving from the side direction is higher than for those coming from that of front or rear directions. This may be because side-entering particles produce clearer projections in the images, providing the model with more distinguishable features and thereby improving classification accuracy. In contrast, particles arriving from the front or rear exhibit greater variability in their image representation due to differences in viewing angles
and distances (as shown in Fig. 13), leading to a reduction in classification accuracy. In summary, the proposed method demonstrates a certain degree of robustness against wind, particularly when wind speed is below 5 m/s, where it continues to perform reliably and effectively mitigates the impact of wind on precipitation particle classification.





In addition to the discussion on algorithm accuracy, computational complexity is also a critical factor in practical applications. Specifically, when scaling from a single camera to a large-scale surveillance network, the overall computational complexity may increase exponentially, significantly impacting system efficiency and resource consumption. Therefore, evaluating and optimizing the algorithm's computational cost while maintaining identification accuracy is essential for ensuring feasibility in real-world deployments. Here, two crucial metrics for evaluating the complexity and practicality of deep learning models for GHP identification are Floating Point Operations (FLOPs) and Parameters(Rump, Ogita and Oishi 2008, Carr and Kennedy 1994). FLOPs denote the total number of floating-point operations required to execute a network model once, reflecting the computational demand during a single forward propagation. This metric is widely used to assess a model's computational efficiency and processing speed. Meanwhile, Parameters encompass the total number of parameters within a model, as well as those that require training, which indicates the GPU memory resources needed for model training. A detailed comparison of FLOPs and Parameters across various deep-learning models is presented below.

**Table 9: The Parameters of deep-learning models.**

| | | RNN | LSTM | GRU | 1D-CNN | Bi-LSTM |
|---|---|---|---|---|---|---|
| DenseNet 121 | Total | 7048781 | 7048811 | 7048801 | 7048785 | 7048803 |
| | Trainable | 6965133 | 6965163 | 6965153 | 6965137 | 6965155 |
| EfficientNet B0 | Total | 4063657 | 4036387 | 4063677 | 4063661 | 4063679 |
| | Trainable | 4021641 | 4021671 | 4021661 | 4021645 | 4021663 |
| Inception V3 | Total | 21825325 | 21825355 | 21825345 | 21825329 | 21825347 |
| | Trainable | 21790893 | 21790923 | 21790913 | 21790897 | 21790915 |
| ResNet 50 | Total | 23610253 | 23610283 | 23610273 | 23610257 | 23610275 |
| | Trainable | 23557133 | 23557163 | 23557153 | 23557137 | 23557155 |
| MobileNet V2 | Total | **2272077** | **2272107** | **2272097** | **2272081** | **2272099** |
| | Trainable | **2237965** | **2237995** | **2237985** | **2237969** | **2237987** |

**Table 10: The FLOPs of deep-learning models.**

| | RNN | LSTM | GRU | 1D-CNN | Bi-LSTM |
|---|---|---|---|---|---|
| DenseNet 121 | 30213894 | 30213918 | 30213910 | 30213877 | 30213927 |
| EfficientNet B0 | 17203088 | 17203112 | 17203104 | 17203071 | 17203121 |
| Inception V3 | 88144876 | 88144900 | 88144892 | 88144859 | 88144909 |
| ResNet 50 | 95565102 | 95565126 | 95565118 | 95565085 | 95565135 |
| MobileNet V2 | **9880644** | **9880668** | **9880660** | **9880627** | **9880677** |





As shown in Tables 9 and 10, our proposed method exhibits significantly lower Parameters and FLOPs values compared to deep learning models based on alternative spatial feature extraction frameworks. While the GRU, which is used as our

temporal feature extraction framework, presents a slightly higher complexity than RNN, LSTM, and 1D-CNN, it offers a clear advantage in terms of accuracy. This increased complexity is offset by the improved performance, demonstrating the ability of GRU to better capture the temporal dependencies inherent in GHP observation tasks. In summary, the proposed method represents an optimal choice for large-scale deployment and GHP observation applications. It not only achieves superior accuracy but also ensures efficiency in both the HSV dataset and real-world experiments, outperforming other

algorithms in terms of both computational resource usage and recognition performance. This makes it highly suitable for practical, large-scale applications where both accuracy and efficiency are paramount.

## 4.6. Discussion

In our years of video data collection and real-world experiments, we have found that under certain conditions, our method may fail. For example,

● when raindrops adhere to the camera lens, the image becomes blurred, which affects the image quality and leads to inaccurate GHP identification. Since surveillance cameras are typically exposed to the external environment, this issue occurs not only during the day but also at night, as shown in Fig.15 (a) and (b). In particular, under windy conditions, raindrops are more likely to attach to the lens, increasing the blurriness and unclear nature of the image. This not only affects the resolution of precipitation particles but also makes it difficult to accurately classify GHPs.

● Strong winds may also cause camera shake, blurring precipitation images in the surveillance field of view. Under strong wind conditions, the movement trajectories of precipitation particles become unstable, and rain droplets, snowflakes, and other particles may be scattered by the wind, as shown in Fig.15 (c). This not only alters their fall paths but may also cause the precipitation patterns to become unclear, increasing the complexity of algorithmic interpretation.

● High air humidity during precipitation events is also a contributing factor. When humidity increases, particularly during continuous rainfall or wet weather, water droplets or mist tend to condense on the camera lens, leading to blurred images, as shown in Fig.15 (d). This phenomenon is commonly observed during early mornings or at night when humidity levels are higher, and it may also occur during sudden precipitation events. The accumulation of moisture prevents the lens from clearly presenting the shape and movement trajectories of precipitation particles,

further complicating the identification process.

● Lightning can also affect the performance of surveillance cameras. As shown in Fig.15 (e), the intense light from lightning and the rapidly changing environmental conditions can interfere with the camera's automatic exposure system, leading to overexposure or underexposure, or even uneven exposure in the image. This strong light and the rapid changes in the scene can disturb the normal functioning of the algorithm, resulting in misjudgment or loss of

precipitation images, especially in thunderstorms with frequent lightning.





- Additionally, dust on the lens can also affect image quality, though this impact is smaller compared to raindrops or humidity. When dust accumulates on the lens, the image may become slightly blurry, but it won't cause significant distortion like raindrops or fog, as shown in Fig.15 (f). However, in cases of severe dust accumulation, it may affect the separation of GHP from the background, thus impacting the accurate recognition of GHP.

In practical applications, manually cleaning each camera lens is resource-intensive and difficult to implement on a large scale, particularly in large-scale surveillance networks. Currently, advanced image denoising and deblurring techniques have been developed in the field of computer vision, which can improve image quality to some extent by removing blur and enhancing the clarity of surveillance footage(Wang et al. 2020, Li et al. 2021). However, these techniques are primarily designed for conventional monitoring tasks, especially for object detection, such as monitoring people, vehicles, and other

targets. In these applications, precipitation particle images are often considered "noise," causing details and shapes of the precipitation particles to become blurred or even completely lost. This loss of information is critical for particle phase classification, which negatively impacts GHP recognition tasks. To address this issue, two feasible solutions are proposed:

Develop a dedicated video/image quality recognition model: This model could evaluate image clarity and identify abnormal images caused by raindrop attachment, lens blur, high humidity, and other factors leading to degraded image quality. When

the system detects that the image quality is insufficient, it can discard low-quality images. The main function of this model would be to preprocess the input videos or images, determining whether their quality is clear enough to meet the requirements of precipitation phase identification tasks.

Incorporate low-quality images as a new class in the training dataset: By adding low-quality images as a new category, the model can learn how to recognize quality issues in precipitation images and make corresponding judgments. Specifically,

this new class could be labelled as "low-quality image" or a similar label, representing images that are affected by raindrops, mist, lens stains, or other factors that degrade their quality. In this way, the model can not only recognize normal precipitation phases but also effectively differentiate which images cannot be accurately classified due to quality issues, thereby improving the accuracy and reliability of the results.

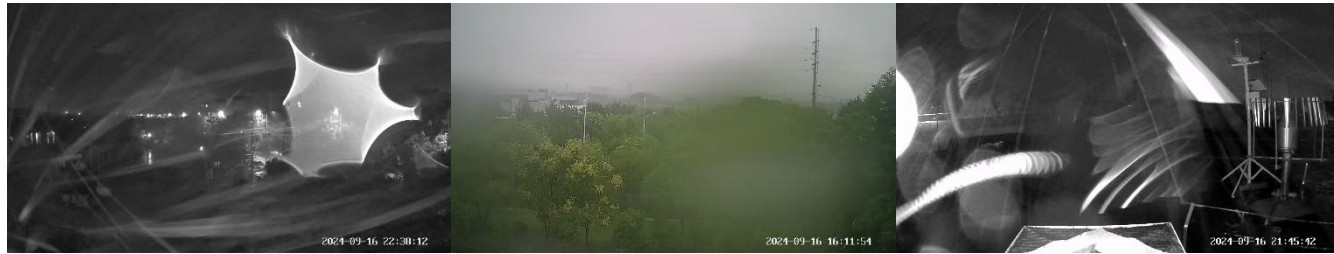

(a) Drop attachment on lens at night     (b) Daytime drop attachment on lens     (c) Wind caused image blurred





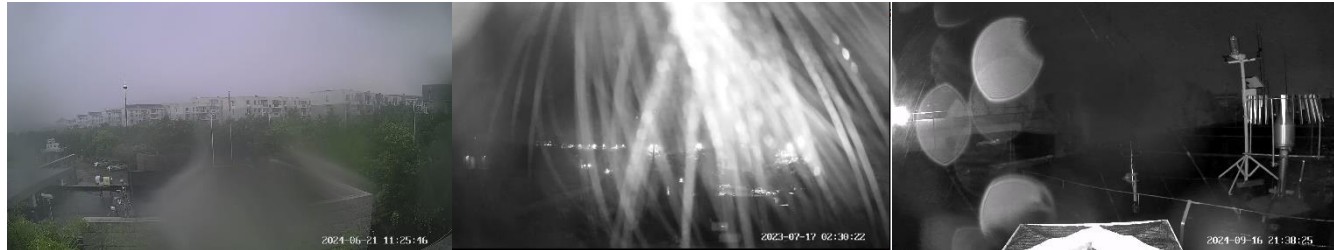

(d) Mist caused lens blurred     (e) Lightning-induced exposure anomalies   (f) Dust caused image blurred

**Figure 15: Different types of surveillance image quality degradation.**

## 5. Conclusion

In this study, we focus on identifying three hydrometeor phase states—rainy, snowy, and graupel—using surveillance cameras. We analyse their distinguishing characteristics in both daytime and nighttime videos to inform our classification approach. To balance precision, latency, and efficiency requirements in real-world applications, we employ a MobileNet V2 network with transfer learning to extract image and spatial features, followed by a GRU network to capture temporal information, enabling high-accuracy GHP discrimination. For training and testing, we constructed the HSV dataset, a GHP
video dataset totalling approximately 94 hours. To evaluate the performance of our proposed deep learning model, we compared it against 24 alternative deep-learning models. Experiments on the HSV dataset show that the proposed algorithm achieves an optimal accuracy of 0.9677 when NFS = 10. Although some comparative algorithms demonstrate slightly lower accuracy, our method exhibits significantly reduced computational and time complexity, making it highly suitable for practical deployment. Furthermore, six real-world experiments yielded an average accuracy of 0.9301, with comparable
performance during both daytime and nighttime, demonstrating the algorithm's stability even when faced with varying camera parameters. Moreover, our method demonstrates a certain degree of wind resistance, achieving satisfactory performance when wind speed is below 5 m/s. This robustness makes our method a viable solution for large-scale, all-day, high-accuracy GHP observation tasks.

Currently, the method faces limitations in distinguishing between rain and graupel, with the recognition accuracy for graupel
reaching only 0.8726 in real-world applications. Enhancing graupel discrimination accuracy is a key area for future improvement. Additionally, addressing challenges such as reducing misclassifications in "no precipitation" conditions and improving the system's ability to detect failure cases in special scenarios will be essential for increasing reliability and applicability in diverse real-world environments. To further enhance the recognition of various GHPs, the HSV dataset will be expanded to include hail surveillance videos.


*Competing interests.*

The contact author has declared that none of the authors has any competing interests.





*Acknowledgment.*

This research was funded by the National Natural Science Foundation of China (NSFC) (No.42405140 and 42025501), the China Postdoctoral Foundation (No. 2024M761383), the Fundamental Research Funds for the Central Universities—Cemac "GeoX" Interdisciplinary Program (No. 020714380210), the Open Grants of China Meteorological Administration Radar Meteorology Key Laboratory (No. 2024LRM-A01 and 2024LRM-A02), and the Talent Startup project of NJIT (YKJ.202315).


*Data availability.*

Due to the sensitivity of urban surveillance data, partially processed (masking of people and cars in the video) experimental surveillance videos are available at: https://pan.baidu.com/s/102PeNAcsi1NdA1Bd9AW70A (code: CPPD). Examples of rain, snow, and graupel in visible and near-infrared videos (in GIF format) can be obtained from the:

https://pan.baidu.com/s/1yLnDzD3Vmd4x6iHdVizP0A?pwd=cppd (code: cppd).

## Appendix. A

The Confusion matrixs of different deep learning algorithms are presented as follows:

**Table A1: Confusion matrix of different deep learning algorithms (NFS = 15).**

| RNN | LSTM | GRU | 1D-CNN | Bi-LSTM |
|---|---|---|---|---|

**RNN (top row):**

| True Label \ Predicted | No_P | Rainy | Snowy | Graupel |
|---|---|---|---|---|
| Graupel | 1369 | 38 | 37 | 26 |
| Snowy | 8 | 2032 | 41 | 65 |
| Rainy | 17 | 68 | 1488 | 24 |
| No_P | 34 | 105 | 36 | 1381 |

**LSTM (top row):**

| True Label \ Predicted | No_P | Rainy | Snowy | Graupel |
|---|---|---|---|---|
| Graupel | 1351 | 73 | 110 | 162 |
| Snowy | 21 | 1955 | 23 | 25 |
| Rainy | 32 | 99 | 1375 | 34 |
| No_P | 24 | 116 | 94 | 1275 |

**GRU (top row):**

| True Label \ Predicted | No_P | Rainy | Snowy | Graupel |
|---|---|---|---|---|
| Graupel | 1345 | 21 | 18 | 16 |
| Snowy | 22 | 2135 | 30 | 25 |
| Rainy | 17 | 14 | 1525 | 33 |
| No_P | 44 | 73 | 29 | 1422 |

**1D-CNN (top row):**

| True Label \ Predicted | No_P | Rainy | Snowy | Graupel |
|---|---|---|---|---|
| Graupel | 1281 | 5 | 59 | 47 |
| Snowy | 14 | 2077 | 31 | 121 |
| Rainy | 32 | 10 | 1492 | 18 |
| No_P | 101 | 159 | 20 | 1310 |

**Bi-LSTM (top row):**

| True Label \ Predicted | No_P | Rainy | Snowy | Graupel |
|---|---|---|---|---|
| Graupel | 1295 | 50 | 19 | 22 |
| Snowy | 48 | 2075 | 49 | 58 |
| Rainy | 36 | 45 | 1514 | 26 |
| No_P | 49 | 73 | 20 | 1390 |

**RNN (middle row):**

| True Label \ Predicted | No_P | Rainy | Snowy | Graupel |
|---|---|---|---|---|
| Graupel | 1228 | 61 | 43 | 41 |
| Snowy | 58 | 2038 | 36 | 85 |
| Rainy | 77 | 42 | 1048 | 20 |
| No_P | 65 | 102 | 75 | 1350 |

**LSTM (middle row):**

| True Label \ Predicted | No_P | Rainy | Snowy | Graupel |
|---|---|---|---|---|
| Graupel | 1285 | 67 | 28 | 37 |
| Snowy | 43 | 2001 | 30 | 90 |
| Rainy | 31 | 53 | 1512 | 22 |
| No_P | 69 | 122 | 32 | 1437 |

**GRU (middle row):**

| True Label \ Predicted | No_P | Rainy | Snowy | Graupel |
|---|---|---|---|---|
| Graupel | 1397 | 56 | 27 | 29 |
| Snowy | 14 | 2035 | 30 | 67 |
| Rainy | 9 | 45 | 1502 | 46 |
| No_P | 8 | 107 | 43 | 1354 |

**1D-CNN (middle row):**

| True Label \ Predicted | No_P | Rainy | Snowy | Graupel |
|---|---|---|---|---|
| Graupel | 1309 | 72 | 45 | 82 |
| Snowy | 54 | 2043 | 43 | 107 |
| Rainy | 46 | 30 | 1493 | 42 |
| No_P | 19 | 98 | 21 | 1265 |

**Bi-LSTM (middle row):**

| True Label \ Predicted | No_P | Rainy | Snowy | Graupel |
|---|---|---|---|---|
| Graupel | 1311 | 47 | 34 | 11 |
| Snowy | 38 | 2080 | 45 | 67 |
| Rainy | 28 | 34 | 1486 | 18 |
| No_P | 51 | 82 | 37 | 1400 |

**RNN (bottom row):**

| True Label \ Predicted | No_P | Rainy | Snowy | Graupel |
|---|---|---|---|---|
| Graupel | 1321 | 78 | 37 | 27 |
| Snowy | 46 | 2018 | 42 | 89 |
| Rainy | 28 | 53 | 1484 | 36 |
| No_P | 33 | 94 | 39 | 1344 |

**LSTM (bottom row):**

| True Label \ Predicted | No_P | Rainy | Snowy | Graupel |
|---|---|---|---|---|
| Graupel | 1311 | 34 | 110 | 3 |
| Snowy | 30 | 2101 | 12 | 83 |
| Rainy | 66 | 7 | 1475 | 14 |
| No_P | 21 | 101 | 5 | 1396 |

**GRU (bottom row):**

| True Label \ Predicted | No_P | Rainy | Snowy | Graupel |
|---|---|---|---|---|
| Graupel | 1303 | 72 | 35 | 14 |
| Snowy | 41 | 2022 | 42 | 48 |
| Rainy | 52 | 43 | 1487 | 16 |
| No_P | 32 | 106 | 38 | 1418 |

**1D-CNN (bottom row):**

| True Label \ Predicted | No_P | Rainy | Snowy | Graupel |
|---|---|---|---|---|
| Graupel | 1357 | 70 | 17 | 75 |
| Snowy | 28 | 2075 | 8 | 75 |
| Rainy | 3 | 38 | 1511 | 25 |
| No_P | 40 | 60 | 66 | 1321 |

**Bi-LSTM (bottom row):**

| True Label \ Predicted | No_P | Rainy | Snowy | Graupel |
|---|---|---|---|---|
| Graupel | 1367 | 52 | 30 | 21 |
| Snowy | 17 | 2030 | 18 | 52 |
| Rainy | 16 | 56 | 1536 | 16 |
| No_P | 28 | 105 | 18 | 1407 |





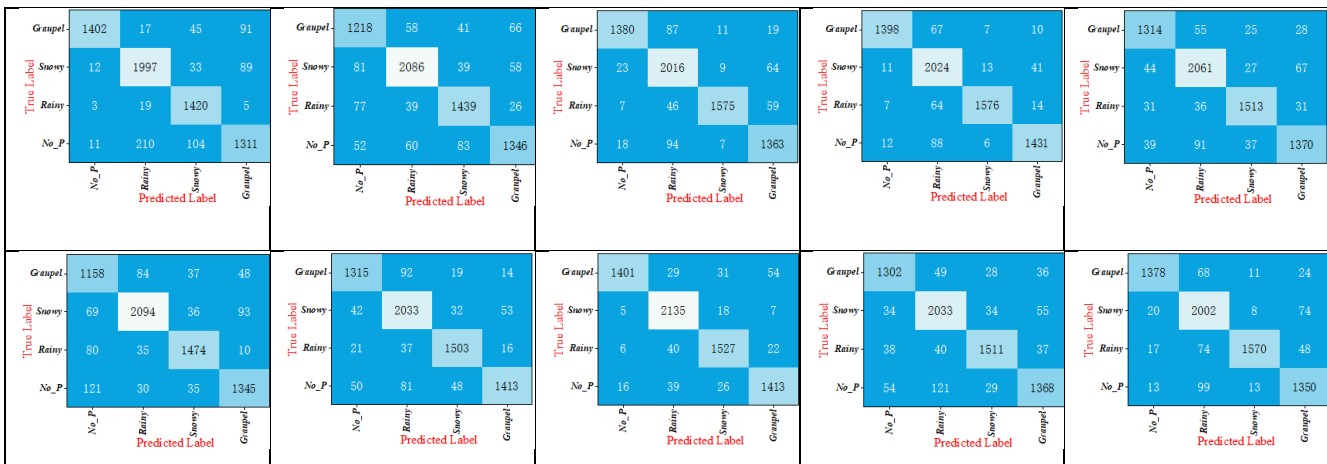

**Note:** First row: DenseNet 121; Second row: EfficientNet B0; Third row: Inception V3; Forth row: ResNet 50; Fifth Row: MobileNet V2.

**Table A2: Confusion matrix of different deep learning algorithms (NFS = 10).**

| RNN | LSTM | GRU | 1D-CNN | Bi-LSTM |
|---|---|---|---|---|





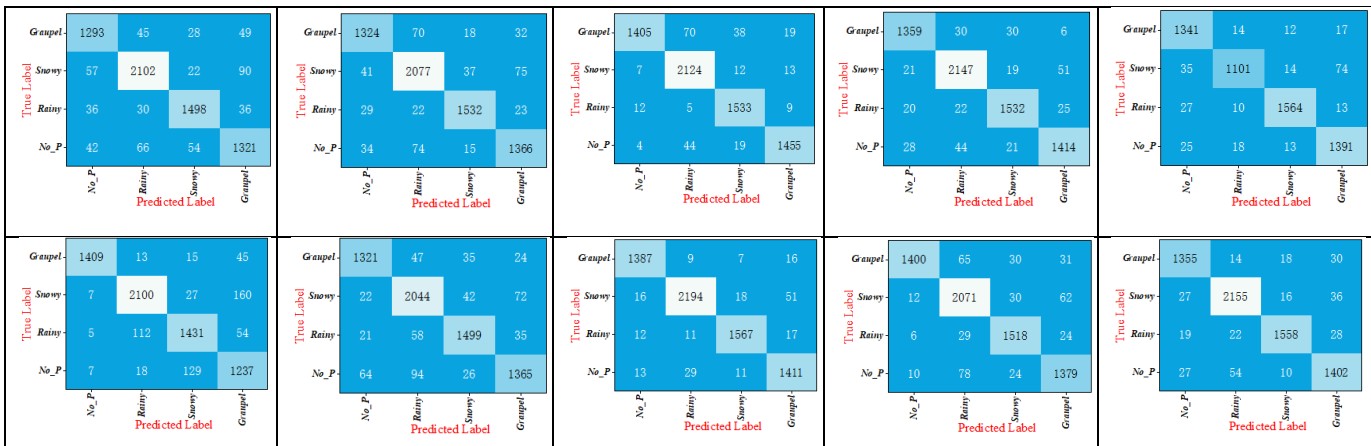

**Note:** First row: DenseNet 121; Second row: EfficientNet B0; Third row: Inception V3; Forth row: ResNet 50; Fifth Row: MobileNet V2.

**Table A3: Confusion matrix of different deep learning algorithms (NFS = 5).**

| RNN | LSTM | GRU | 1D-CNN | Bi-LSTM |
|---|---|---|---|---|





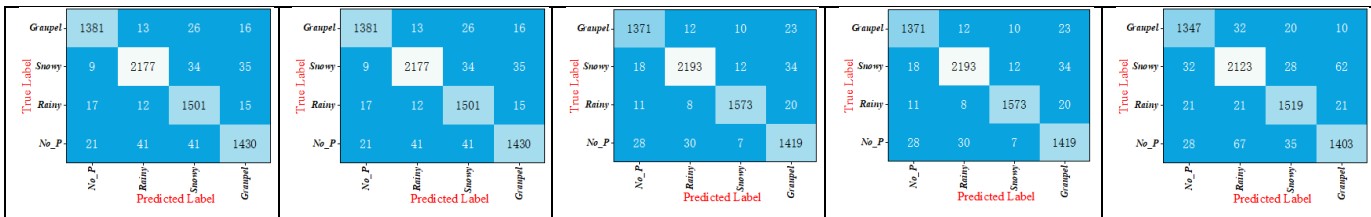

**Note:** First row: DenseNet 121; Second row: EfficientNet B0; Third row: Inception V3; Forth row: ResNet 50; Fifth Row: MobileNet V2.


## Appendix. B

Real-world GHP identification by surveillance camera_2 and camera_3 is shown as below:

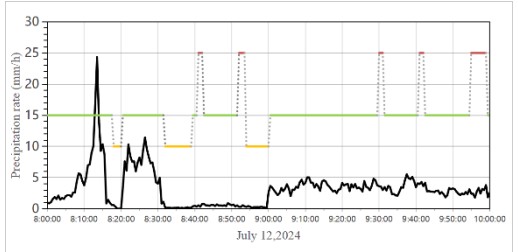

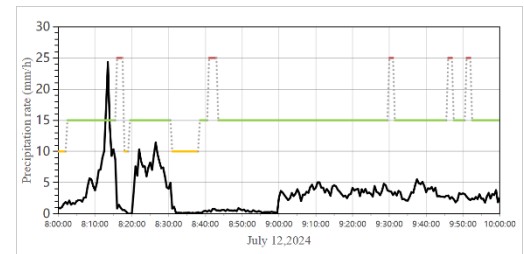

**(a) Day-time Rain of camera_2**  **(b) Day-time Rain of camera_3**

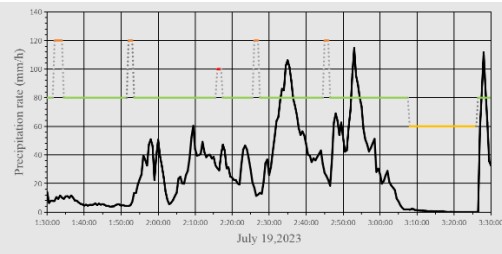

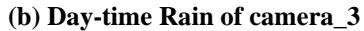

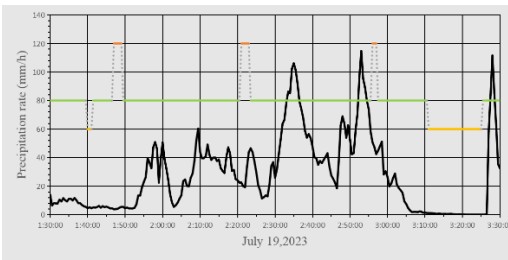


**(c) Night-time Rain of camera_2**  **(d) Night-time Rain of camera_3**

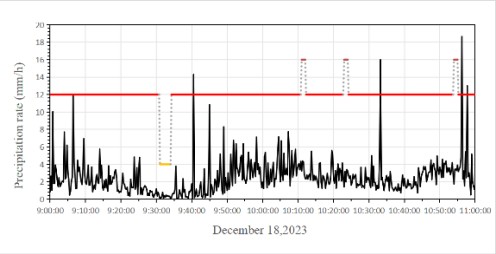

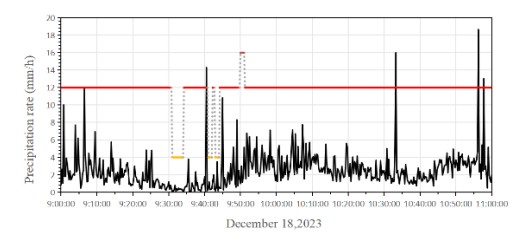

**(e) Day-time snow of camera_2**  **(f) Day-time snow of camera_3**





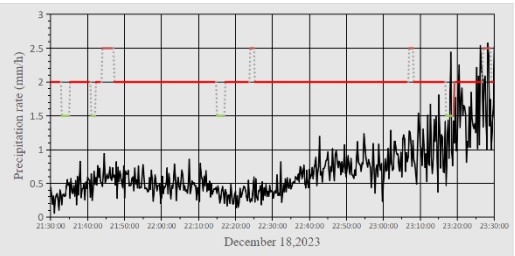
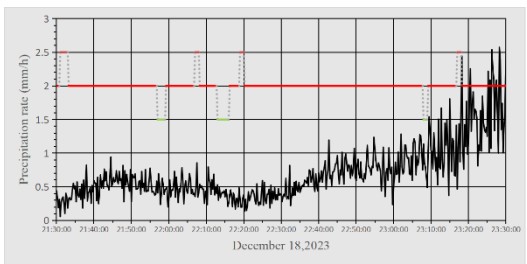

**(g) Night-time snow of camera_2**          **(h) Night-time snow of camera_3**

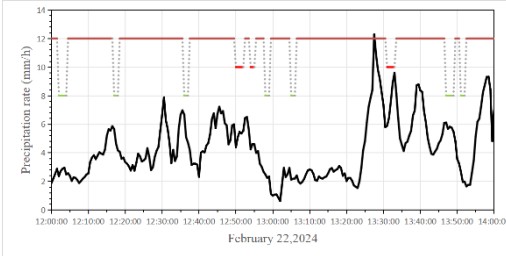
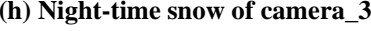
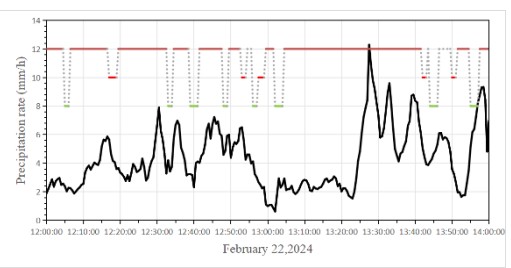

**(i) Day-time graupel of camera_2**          **(j) Day-time graupel of camera_3**

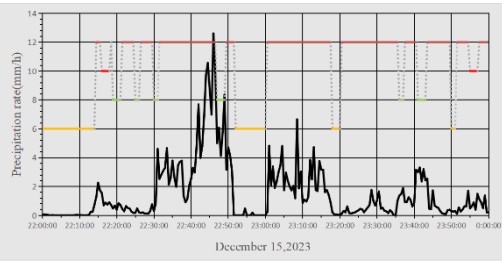
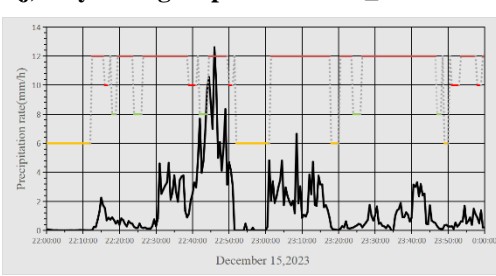

**(k) Night-time graupel of camera_2**          **(l) Night-time graupel of camera_3**

**Figure B1: Real-world GHP identification by surveillance camera_2 and camera_3.**

**( ▬▬▬ : rainy; ▬▬▬ : snowy; ▬▬▬ :graupel; ▬▬▬ : no precipitation; The black curve represents the precipitation intensity readings from the 2-DVD)**

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
