# Peer review of "Surveillance Camera-Based Deep Learning Framework for High-Resolution Ground Hydrometeor Phase Observation"

_Atmospheric Measurement Techniques, 2024_

## Author Comment (AC1)

**Reply on RC1**

Dear Editor and Reviewers,

Thank you for the comments.
We gratefully thank the editor and all reviewers for their time spent making their constructive remarks and useful suggestions, which has significantly raised the quality of the paper and has enabled us to improve the paper. Each suggested revision and comment brought forward by the reviewers was considered and incorporated. We will be happy to edit the text further based on helpful comments from the reviewers.
The text style for revision is as follows:

- All comments are in **black**.
- All responses to the comments are in **blue**.
- All pages and line numbers refer to the revised manuscript.
- All revised contents in the manuscript are highlighted in **red**.

General Comments

The manuscript describes a method, based on the analysis of surveillance camera observations, to estimate surface precipitation type conditions. Overall the topic is well suited to AMT and the methodology seems also adequate for the purpose of the study. However, I think there are substantial clarifications and some corrections to be made to further consider this manuscript for publication.

First the text should clarify that the focus is the precipitation type falling at ground level, i.e. close to the surface. In some parts seems that ground level conditions, i.e. snow on ground, is also considered.

Reply:

Thank you for your reminder. Our method primarily focuses on the process of precipitation particles falling, as captured by surveillance video, rather than the snow or water accumulation on the ground. We have emphasized this point in multiple sections of the revised manuscript based on your suggestions.

Second, authors should check carefully the terminology used, particularly considering that the journal is specialized in meteorological observational techniques. A precise terminology is essential to avoid confusions. For example, regarding the 3 hydrometeors considered in the analysis (for example in Fig. 3), rain, snow and

graupel, I'm wondering if graupel is really considered or they mean a mixture of snow and rain, as other studies which simplify the wide variety of hydrometeor types considering only 3 precipitation phase types: liquid, solid and mixed. I include below some references to studies mentioning graupel, which in general, is far less frequent than mixed (solid and liquid) phase types (of course other studies in the literature can also be considered).

Reply:

We also greatly appreciate the recommended references.

It is important to emphasize that in this study, graupel refers specifically to the solid state, and the mixed-phase state is not considered. Therefore, if we refer to the classification methods in existing studies, our study considers the solid and liquid states, but the mixed-phase state is not included. We have highlighted this point in the article, as seen in line 226-227.

Finally, a number of formal corrections, language checking, etc. should also be performed in some parts. I indicate below some items as a reference but I don't intend to be exhaustive here.For all the above I think major reviews are necessary to improve the current manuscript.

Specific Comments

1. Page 1, title, line 10-11 and elsewhere. When first used please clarify the meaning of GHP, refered to near surface precipitation type, not to ground conditions.

2. Page 1, line 28. Suggest: Hydrometeors -> Precipitating hydrometeors

3. Page 2, line 37. Suggest: rainfall phase -> liquid precipitation / solid snowfall -> solid precipitation

4. Page 2, line 38. Please reconsider: interchange -> alternate?

Reply:

The authors have accepted the above suggestions and made the necessary revisions in the revised manuscript.

5. Page 3, last line and elsewhere. Suggest looking for an alternative term to 'generalized visual data'

Reply:

Thank you for your reminder. We have replaced " user-generated visual data " with " generalized visual data."

6.  Page 4, first paragraph: 'is primarily attractive to professional meteorological researchers'. I think the AMT journal audience could be defined generally as 'professional meteorological researchers' so I would rewrite this paragraph accordingly. I think in this journal the term 'hydrometeor' should be preferred if you consider only precipitation phase partitioning. If you want to include precipitating hydrometeors and other phenomena such as haze or fog then perhaps you can use 'weather conditions'.

Reply:
Thank you for your valuable feedback. We appreciate your suggestion to revise the phrase "is primarily attractive to professional meteorological researchers." We agree that the AMT journal's audience can be broadly defined as "professional meteorological researchers" and will modify the wording to reflect that. Please see lines 93-95.

7.  Page 4, section 2.1. Authors should clarify if "indirect measurements' are intended to provide precipitating hydrometeor types or ground conditions: for example it is not the same to distinguish if there's snow on the ground or if it's snowing.

8.  Page 5, line 130. Haze is not a precipitating hydrometeor.

9.  Page 5, line 133. Suggest: weather -> weather conditions

10. Page 6, line 162. Suggest: includes -> may include

11. Page 8, Table 1. The term 'sandy' does it mean that you recognize 'ground', 'bare soil' or it means really 'sandy' (as in the case of a beach).

12. Page 9, line 226: shrapnel particles refer to graupel particles? This term was not used before, unlike graupel.

13. Page 11, beginning of last paragraph. 'Figure 6' refers to Figure 5? Please check.

14. Page 13. Please provide reference(s) for terminal velocity formulas used for different hydrometeors.

15. Page 14, line 342. The rainfall rate values given (in mm/h) were recorded over which time periods, hourly? Note that it is not the same 195 mm/h during 10 min than during 1 h.

16. Page 15, Table 2. Please indicate in the table the units used of the values listed. Are they events of different time duration?

17. Page 18, Table 6. Please indicate the meaning of values in bold, best scores?

18. Page 22, Table 8. Which score is used in the table? Please indicate explicitly in the table title.

19. Page 26, Table 9 and 10. Please indicate meaning of bold and underscored values.

Reply: Thank you for your suggestion.

For Question 7, we have added the necessary information, please see lines 107-108.

For Question 13, after checking, we confirm that Figure 6 is correct.

Question 15, the precipitation intensity values are calculated every minute. Therefore, the values in Table 2 refers to the precipitation intensity during a 1-minute period, not an hourly intensity. Please see 373-375.

Question 16, Yes, these video data was collected during a long period of observation (starting from March 2023 and ending in July 2024). (please see lines)

We have accepted the other suggestions provided by the reviewer and made the corresponding revisions. Please see the revised manuscript.

Technical Comments

1. Page 1, line 29 (and elsewhere). Please check citation style, shouldn't it be Pruppacher et al. (1998) (when there are more than two authors et al. should be used)?

2. Page 3, lines 74-75: duplicate: 'during daytime and nightime'

3. Page 3, line 86: as follows: Following -> as follows. Following

4. Page 4, line 111. Typo: labeled -> label

5. Page 5, line 127. Please rephrase: have the disadvantage of needing to be quicker

6. Page 7, and elsewhere: Table.1 -> Table 1

7. Page 8, Table 1. Please rewrite the references: (Zhao et al. 2011) -> Zhao et al. (2011), etc.

8. Page 8, line 221 (and page 9, line 234). Suggest: graupel -> graupel particles

9. Page 9, line 235. Correct : rain -> rain drops

10. Page 9, line 240. Please check meaning and correct : rain -> raindrop trajectories

11. Page 13, after equation 4 (and elsewhere after other equations). Where -> where [in lower case]

12. Page 16, line 387. Correct: follows -> listed in Table 3.

13. Page 18, line 419 (and also Fig. 10 caption). Suggest: a violin plot … quantifies -> violin plots … quantify

14. Page 21-22, Fig. 12. X-axis labels hard to read.

15. Page 23, line 504. Do you mean: As analyzed in Section 3?

16. Page 25, please check the size of the delta symbol.

17. Page 28, line 628. Typo: please add blank space after 'footage'

Reply:

We would like to express our sincere gratitude to the reviewer for their insightful comments. The above issues have all been thoroughly addressed and corrected. We have carefully reviewed the relevant sections and made the necessary adjustments to ensure accuracy and clarity. Your feedback has been invaluable in improving the quality of our work.

---

## Author Comment (AC2)

**Reply on RC2**

Dear Editor and Reviewers,

Thank you for the comments.
We gratefully thank the editor and all reviewers for their time spent making their constructive remarks and useful suggestions, which has significantly raised the quality of the paper and has enabled us to improve the paper. Each suggested revision and comment brought forward by the reviewers was considered and incorporated. We will be happy to edit the text further based on helpful comments from the reviewers.
The text style for revision is as follows:

- All comments are in **black**.
- All responses to the comments are in **blue**.
- All pages and line numbers refer to the revised manuscript.
- All revised contents in the manuscript are highlighted in **red**.

1. This paper uses a camera system to detect hydrometeor types such as RN, SN, and GR. Uses ML techniques but not clear how it is trained using observed input parameters.

Reply:

Thanks for the comments.

The surveillance video is divided into 5-second segments. Within each segment, sequences of 5, 10, and 15 frames are selected and fed into the spatial feature extraction module (the effect of sequence length on the classification results is discussed in Section 4.4). The extracted feature vectors are then input into the temporal model for precipitation type classification. We have added these description in lines 269-277.

2. Camera systems can provide yes or no question on RN, SN, or FG detection (see Gultepe et al 2009AMS Bull, AMS Monographs on Solid precipitation 2017; AMS Bull Ice fog (2014). But doesnt provide any other info on particle shape, size, and cncentration.

Reply:

Thanks for the comments.

Based on the authors' understanding of surveillance camera imaging principles, researchers can extract information such as droplet size, droplet spectra, and other microphysical characteristics of precipitation, provided they have access to the necessary camera parameters.

For example, the following studies demonstrate how cameras can be employed to calculate the Drop Size Distribution and other microphysical properties of precipitation.

- Allamano, P., Croci, A. and Laio, F., 2015. Toward the camera rain gauge. Water Resources Research, 51(3): 1744-1757.
- Jiang, S., Babovic, V., Zheng, Y. and Xiong, J., 2019. Advancing opportunistic sensing in hydrology: A novel approach to measuring rainfall with ordinary surveillance cameras. Water Resources Research, 55(4): 3004-3027.
- Wang, X., Wang, M., Liu, X., Zhu, L., Glade, T., Chen, M., Zhao, W. and Xie, Y., 2022. A novel quality control model of rainfall estimation with videos–A survey based on multi-surveillance cameras. Journal of Hydrology, 605: 127312.

The above three studies primarily focus on precipitation captured by daytime visible-light surveillance videos, whereas the following study specifically examines nighttime precipitation.

- Lee, J., Byun, J., Baik, J., Jun, C. and Kim, H.-J., 2022. Estimation of raindrop size distribution and rain rate with infrared surveillance camera in dark conditions. Atmospheric Measurement Techniques Discussions: 1-23.

3. Looking at the only visible images cant resolve the particle discrimination issue.

Reply:

Thanks for the comments.

Please refer to the response to Question2.

Therefore, we attempt to distinguish precipitation phases using image-based analysis. This approach is based on the fact that different precipitation types exhibit distinct visual characteristics in surveillance footage, such as variations in brightness, shape, size, and motion patterns. By leveraging these features, we aim to classify precipitation phases more effectively.

4. Visually helps what is on the ground if Visibility lows but many times this cant be true because during precip Vis goes down.

Reply:

Thanks for the comments.

The increase in precipitation intensity and variations in ambient lighting conditions does affect visibility (as analyzed in section 4.6). It is important to clarify that surveillance cameras capture visible-light videos during the daytime, while at night or under low-light conditions, they record near-infrared videos to achieve imaging in the monitored area, enabling "night vision" functionality.

As shown in Figure 12, our experiments cover various precipitation scenarios from daytime to nighttime and across different intensities. These include conditions ranging from drizzle to heavy rain with an intensity of up to 110 mm/h, from light snow to blizzards with a precipitation intensity of approximately 20 mm/h, and graupel events with an intensity of 12 mm/h. As discussed in lines 533–542, we have compared the performance of our method under these conditions. Experimental results indicate that while performance varies under different visibility conditions, the proposed method remains generally stable overall.

5. This is well know that Vis doesnt provide precip type or amount unless you know the particle type. In reality, all your analysis is based on visibility of light reflected/scatterered due to hydrometeors.

Reply:

Thanks for the comments.

In response, the authors Need evidence that our method leverages the reflection and scattering of light by precipitation particles, which manifest in surveillance videos as differences in particle shape, color, and brightness—key spatial features for distinguishing precipitation types. Additionally, we utilize the falling speed of particles, as meteorological studies have established that different particles exhibit varying fall velocities. As described in Section 3.1.2 on the temporal feature extraction module, we enhance the accuracy of phase differentiation by incorporating these temporal variations, such as fall speed, alongside the spatial characteristics of precipitation particles.

6. It is clear that we can detect precip for yes or no with help of a Temp probe. Therefore, i feel this manuscript needs to be reduced significantly focusing on the clear objectives. Otherwise, this kind of work can lead to very limited applications.

Reply:

Thanks for the comments.

We appreciate the reviewer's suggestion on the temperature inclusion of our work. Certainly, incorporating temperature measurements could potentially enhance the accuracy of the results, and this is something that could be explored in future work.

However, our study focuses solely on utilizing surveillance video data to distinguish between rain, snow, and graupel through deep learning methods. The primary motivation behind this research is to take advantage of surveillance cameras with their large number, high density, and fast transmission capabilities. Additionally, this approach can be implemented using existing urban surveillance resources, which helps reduce both maintenance and installation costs. Since temperature measurements are not typically available from surveillance cameras, we focus on the visual characteristics of precipitation captured in the videos. This offers a novel solution for various urban environments, providing continuous rain, snow, and graupel monitoring with low deployment costs.

There are several issues in the paper:

1. captions are not clear enough to provide detailed info on figs.

Reply:

We have made careful revisions to the figure captions in the manuscript. Please refer to the revised version.

2. Velociry means what?

Reply:
We did not find the word "Velociry" in our manuscript. We assume you meant "velocity," which in this context refers to the terminal velocity of particles as they fall towards the ground. We have already added this clarification in the revised manuscript.

3. what 2 diff graupel types have very large diffs?

Reply:

We apologize, but we find it difficult to understand the meaning of this sentence. We would greatly appreciate it if you could point out the specific page or line.

We can only speculate that you are referring to the differences in graupel types shown in Figure 7. In this case, we have added relevant references. Figure 7 was primarily created based on meteorological research findings, and we have included additional information in the figure caption to clarify that this study focuses on solid-state graupel particles.

4. Issues with blowing snow and fog are not discussed (Gultepe et al Pure and Appl Geop 2018 Aviation Meteorology) and not provided.

Reply:

Thanks for the comments. We have discussed the impact of wind on precipitation phase differentiation, and snow is, of course, included. Please see lines 555-601.

Fog is outside the scope of our focus, so it was not addressed. However, we have acknowledged the interference caused by fog to our method in the discussion section, as it represents a special case where the method fails to perform effectively. Please see section 4.6.

5. about 90% success in the results to me is too high, it can be ok for yes or no for precip but not the type discrimination.

Reply:
Thank you for your valuable feedback.
We understand your concern regarding the reported 90% success rate in the results. As an example from the articles cited in our paper, Khan et al. (2022) used the ResNet18 architecture and achieved an unprecedented overall detection accuracy of 97% for weather detection. Ibrahim et al. (2019) proposed a new framework called WeatherNet, which achieved an accuracy of over 90%. Xiao et al. (2021) introduced a novel deep CNN, MeteCNN, for weather phenomena classification, which achieved an accuracy of 92%. These studies demonstrate the excellent performance of deep learning methods in weather detection and classification tasks,

While it may seem high for precipitation type discrimination at first glance, the reported accuracy of 90% is consistent with the results of previous studies in the field, such as those by Khan et al. (2022) and Ibrahim et al. (2019), where similar or even higher accuracies were achieved. This level of accuracy is based on a robust dataset

and extensive model fine-tuning. It is important to acknowledge that distinguishing between different types of precipitation is inherently more complex than a simple yes/no classification. The 90% accuracy reflects the overall performance, but additional metrics such as precision, recall, and confusion matrices are provided to give a more nuanced understanding of the model's performance across different precipitation types. Moreover, our method comprehensively integrates both the image features and temporal characteristics of different precipitation types. A detailed analysis of error cases is also included, identifying specific precipitation types where the model's performance may be less reliable, and providing insights into potential reasons for these limitations, such as ambiguities in visual features or challenging environmental conditions. Therefore, achieving this level of accuracy is well-supported by sufficient evidence.

**Reference:**

Khan, M.N., Ahmed, M.M.J.I.j.o.t.s. and technology, 2022. Weather and surface condition detection based on road-side webcams: Application of pre-trained convolutional neural network. 11(3): 468-483.

Ibrahim, M.R., Haworth, J. and Cheng, T.J.I.I.J.o.G.-I., 2019. WeatherNet: Recognising weather and visual conditions from street-level images using deep residual learning. ISPRS International Journal of Geo-Information, 8(12): 549.

Xiao, H., Zhang, F., Shen, Z., Wu, K. and Zhang, J., 2021. Classification of weather phenomenon from images by using deep convolutional neural network. Earth and Space Science, 8(5): e2020EA001604.

5. did you provide a field campaign prediction analysis for particle detection?

Reply:

Thanks for the comments.

We provide a comparative analysis in Section 4.5.

The starting point of our research is to leverage the high spatiotemporal resolution and low-cost observational advantages of urban surveillance cameras. Therefore, real-world observations primarily focus on cameras deployed in urban areas. As shown in Figure 11, the scene captured by Surveillance Camera 2 is more representative of a rural environment, with fewer buildings and less human activity.

6. Finally, definition of direct and indirect is not clear to me. Direct to me if you can measure parameters using insitu sensors. indirect means you get the results based on secondary products.... This needs to be improved.

Reply:
Thank you for raising this issue. In response to a similar suggestion from Reviewer 1, we have addressed this in Section 2.
Direct measurement methods focus on the image/video features exhibited during the precipitation particle's falling process, while indirect measurements involve snow or water accumulation on the ground. For example, indirect measurements focus on whether there is snow on the ground rather than whether it is snowing.

8. Haze related text needs to be improved, abd cant be a hydrometeor type!!! if it is not wet particle.

Reply:

Thank you for the reminder. We have removed the mention of haze.

**Finally, we would like to express our sincere gratitude to your insightful comments and constructive suggestions. Your valuable feedback has greatly contributed to enhancing the clarity and quality of our manuscript, and we truly appreciate the time and effort you dedicated to reviewing our work.**

---

## Author Response (AR2)

**Author Response**

Dear Editor and Reviewers,

Thank you for the comments.

We gratefully thank the editor and all reviewers for their time spent making their constructive remarks and useful suggestions, which has significantly raised the quality of the paper and has enabled us to improve the paper. Each suggested revision and comment brought forward by the reviewers was considered and incorporated. In addition, following the editor's previous comments, the references have been revised to comply with the AMT citation format.

The text style for revision is as follows:

- All comments are in **black**.
- All responses to the comments are in **blue**.
- All pages and line numbers refer to the revised manuscript.

**Reply on RC#1**

**General Comments**

I think the manuscript improved but still needs some minor corrections. Please see below. Specific Comments

- 1. Page 1, section 1, first sentence. Please remove "(not means ground conditions)" and instead add afterwards. "Despite SPT and ground conditions (i.e., snow or rain on ground) are related, note that they are different."
- 2. Page 3, last paragraph: sunny -> sunny or cloudy conditions;
- 3. Page 3, last paragraph: mainly refers to rain, snow, graupel, is primarily attractive to professional meteorological researchers. -> specifically refers here to rain, snow and graupel.
- 4. Page 9, 1st paragraph: Meteorological and physical studies -> Previous meteorological studies
- 5. Page 9, 1st paragraph: graupel refers specifically to the solid state, and the mixed-phase state is not considered -> graupel, also known as snow pellets, refers specifically to solid particles "consisting of crisp, white, opaque ice particles, round

or conical in shape and about 2 - 5 mm in diameter" according to the World Meteorological Organization terminology (WMO, 2017). In this study, no mixed phase precipitation is considered. Reference:WMO 2017, International Cloud Atlas Glossary. Graupel, https://cloudatlas.wmo.int/en/glossary.html#G [last accessed April 2025]

Reply: Authors have accepted the suggestions and revised the manuscript accordingly.

6. Page 14, is Figure 7 quoted in the text? Please check.

Reply: Yes, please see Line 316 of the revised manuscript.

7. Page 14, please provide proper references for each terminal fall speed plotted in Figure 7.

Reply: Added in lines 346-347.

8. Page 15, Figure 8 caption, typo: presents -> present

Reply: Accept and revised.

9. Page 21, Table 7; also in Table 8, Figure 12 and elsewhere, suggest (change adjective into noun): Rainy, Snowy -> Rain, Snow

Reply: Revisions have been made, and Figures 9 and 10 have also been corrected.

**Reply on RC#2**

1. The article introduces an interesting deep-learning based algorithm to identify the precipitation type based on Surveillance camera. The work is timely to support the surge of citizen science for urban precipitation monitoring. I have one major comment for the authors:

By reviewing the manuscript and the authors' response to reviewer 1, it is clear that they did not consider fixed form (solid & liquid) of precipitation in their algorithm. However, fixed form of precipitation is very common, espeially during winter as the authors have addressed in Introduction. It is recommended that the authors include a new category 'unidentified precipitation tyle' in addition to the existing four categories ('rainy', 'snowy', 'graupel', and 'no precipitation'). At minimum, the authors need to explain their choice and its impact on the accuracy of their algorithm.

**Reply:**

Many thanks for your valuable comments and suggestions.

As you have pointed out, mixed-phase precipitation is a very important and commonly occurring form of precipitation in meteorology, especially prominent during winter. Surveillance videos capture precipitation particle groups, and in mixed precipitation scenarios, variations in the proportions of solid and liquid particles lead to significant differences in image and video features. The images of mixed precipitation are not merely a simple superposition of different particle images; optical effects such as refraction and reflection between particles cause the visual features in the videos to differ markedly from those of single-phase precipitation. These optical effects increase the complexity of image features and pose new challenges for modeling the image characteristics of mixed precipitation.

Furthermore, the overall fall velocity of the precipitation particle group in mixed precipitation fluctuates considerably, making it difficult to accurately describe using the theoretical formulas for single-phase precipitation shown in Figure 7. This further increases the difficulty of modeling the spatiotemporal characteristics of precipitation. The current study provides a solid foundation for recognizing mixed-phase precipitation, but the algorithm design is mainly based on microphysical features of single-phase precipitation (such as color, size, fall velocity), which introduces some uncertainty in identifying mixed-phase precipitation. Future work will consider introducing a "mixed precipitation" category or subdividing it into multiple types such as "rain-snow mixture" and "snow-graupel mixture" to more precisely reflect the complexity of actual precipitation processes.

Moreover, limited by the current scarcity of surveillance video samples for mixed precipitation, we plan to further enrich the training and testing datasets, especially by increasing the number of mixed precipitation video samples. By expanding the dataset and optimizing the algorithm, we expect to achieve more accurate and stable recognition of various precipitation types in practical applications, thereby enhancing the model' sutility and potential for wider deployment.

Relevant content has been supplemented in the discussion and conclusion sections of the manuscript. Please see lines 694-708.

We are deeply grateful to the reviewer for their time, thoughtful feedback, and valuable suggestions, which have significantly enhanced the quality and clarity of the manuscript.